# Subspace clustering in high-dimensions: Phase transitions & Statistical-to-Computational gap

**Luca Pesce***
École Polytechnique Fédérale de Lausanne (EPFL)
Information, Learning and Physics lab.

**Bruno Loureiro**
École Polytechnique Fédérale de Lausanne (EPFL)
Information, Learning and Physics lab.

**Florent Krzakala**
École Polytechnique Fédérale de Lausanne (EPFL)
Information, Learning and Physics lab.

**Lenka Zdeborová**
École Polytechnique Fédérale de Lausanne (EPFL)
Statistical Physics of Computation lab.

## Abstract

A simple model to study subspace clustering is the high-dimensional $k$-Gaussian mixture model where the cluster means are sparse vectors. Here we provide an exact asymptotic characterization of the statistically optimal reconstruction error in this model in the high-dimensional regime with extensive sparsity, i.e. when the fraction of non-zero components of the cluster means $\rho$, as well as the ratio $\alpha$ between the number of samples and the dimension are fixed, while the dimension diverges. We identify the information-theoretic threshold below which obtaining a positive correlation with the true cluster means is statistically impossible. Additionally, we investigate the performance of the approximate message passing (AMP) algorithm analyzed via its state evolution, which is conjectured to be optimal among polynomial algorithm for this task. We identify in particular the existence of a statistical-to-computational gap between the algorithm that requires a signal-to-noise ratio $\lambda_{\text{alg}} \geq k/\sqrt{\alpha}$ to perform better than random, and the information theoretic threshold at $\lambda_{\text{it}} \approx \sqrt{-k\rho \log \rho}/\sqrt{\alpha}$. Finally, we discuss the case of sub-extensive sparsity $\rho$ by comparing the performance of the AMP with other sparsity-enhancing algorithms, such as sparse-PCA and diagonal thresholding.

## 1  Introduction

With the growing size of modern data, clustering techniques play an important role in reducing the dimensionality of the features used in modern Machine Learning pipelines. Indeed, in many tasks of interest ranging from DNA sequence analysis to image classification, the relevant features are known to live in a lower-dimensional space (intrinsic dimension) than their raw acquisition format (extrinsic dimension) [1]. In these cases, identifying these features can help saving computational

---

*luca.pesce@epfl.ch

36th Conference on Neural Information Processing Systems (NeurIPS 2022).

resources while significantly improving on learning performance. But given a corrupted embedding of low-dimensional features in a high-dimensional space, is it always *statistically possible* to retrieve them? And if yes - how can reconstruction be achieved *efficiently* in practice? In this manuscript we address these two fundamental questions in a simple model for subspace clustering: a $k$-cluster Gaussian mixture model with sparse centroids. In this model, the low-dimensional hidden features are given by the sparse centroids, which are embedded in a higher dimensional space and corrupted by additive Gaussian noise. We assume that the number of non-zero components of the centroids as well as the number of samples scales linearly with the dimension of the embedding space. Given a finite sample from the mixture, the goal of the statistician is to cluster the data, i.e. estimate the centroids (or features) as well as possible.

## 2 Model & setting

Let $\boldsymbol{x}_\nu \in \mathbb{R}^d$, $\nu \in [n] \coloneqq \{1, \cdots, n\}$ denote $n$ i.i.d. data points drawn from an isotropic $k$-cluster Gaussian mixture:

$$\boldsymbol{x}_\nu \sim_{i.i.d.} \sum_{c \in \mathcal{C}} p_c \mathcal{N}\left(\sqrt{\lambda/s}\boldsymbol{\mu}_c, \mathrm{I}_d\right), \qquad \nu \in [n] \tag{1}$$

where $p_c$ are the class probabilities, $\mathcal{C}$ is the index set ($|\mathcal{C}| = k$), $\boldsymbol{\mu}_c \in \mathbb{R}^d$ are the $s$-*sparse* vectors representing the means of the clusters and $\lambda$ is a measure of the signal-to-noise ratio (SNR) for this model. In the following, we focus in the balanced case for which $p_c = 1/k$. We select the subspace of relevant features thanks to the introduction of the vector $\boldsymbol{v}_i \in \mathbb{R}^k$. The projection of all the cluster means, on a given dimension $i \in [d]$, will be completely determined by a linear combination of the components of $\boldsymbol{v}_i$. We consider for this purpose a Gauss-Bernoulli distribution:

$$\boldsymbol{v}_i \sim_{i.i.d.} \rho \mathcal{N}(0, \mathrm{I}_k) + (1 - \rho)\delta_0 \tag{2}$$

where $\rho \coloneqq s/d$ is the density of non-zero elements. For the convenience of the theoretical analysis that follows, it will be useful to work with a particular encoding of the class labels $\mathcal{C}$. For a given sample $\nu \in [n]$ belonging to the class $c \in \mathcal{C}$, define the following label indicator vector $\boldsymbol{u}_c^\nu \in \{-\frac{1}{k}, \frac{k-1}{k}\}^k$:

$$\boldsymbol{u}_c^\nu = \frac{1}{k}(-1, \ldots, -1, \underbrace{k-1}_{c\text{-th element}}, -1, \ldots, -1) \tag{3}$$

For a given draw $(\boldsymbol{x}_\nu)_{\nu \in [n]}$, define the matrices $\mathrm{X} \in \mathbb{R}^{d \times n}$ with columns given by $\boldsymbol{x}_\nu$ and $\mathrm{U} \in \mathbb{R}^{n \times k}$, $\mathrm{V} \in \mathbb{R}^{d \times k}$ with rows given by $\boldsymbol{u}_c^\nu$ and $\boldsymbol{v}_i$.

A crucial point that we shall exploit in this paper is that, with this notation, our model for subspace clustering can be written as a matrix factorization problem, where the data has been generated as:

$$\mathrm{X} = \sqrt{\frac{\lambda}{s}}\mathrm{VU}^\top + \mathrm{W} \tag{4}$$

with $\mathrm{W} \in \mathbb{R}^{d \times n}$ a Gaussian matrix with elements $W_{i\nu} \sim \mathcal{N}(0, 1)$. This formulation is completely equivalent to the one in eq. (1), by identifying the cluster means as given component-wise by: $\mu_c^{(i)} = \boldsymbol{v}_i^\top \boldsymbol{u}_c$, $i \in [d]$. The centroids will have, in expectation, only $s$ non-zero components; as discussed in the introduction, the $s$-sparse class means $\{\boldsymbol{\mu}_c\}_{c \in \mathcal{C}}$ can be thought of as low-dimensional features embedded in a higher-dimensional space $\mathbb{R}^d$, that have been corrupted by isotropic additive Gaussian noise with variance $\sim \lambda^{-1}$. Given a finite draw $(\boldsymbol{x}_\nu)_{\nu \in [n]}$ generated from this model with class means $\mathrm{V}_\star$ and labels $\mathrm{U}_\star$, the goal of the statistician is to perform *clustering*, i.e. to estimate $\mathrm{U}_\star$ from X, which is equivalent to retrieving the class label for each sample in the data. However, note that there is a clear class symmetry in this problem: the labels can only be estimated up to a permutation $\pi \in \mathrm{Sym}(\mathcal{C})$ reshuffling the columns of $\mathrm{U}_\star$. Taking this symmetry into account we can assess the performance of an estimator $\hat{\mathrm{U}}$ through the averaged symmetrized mean-squared error:

$$\mathrm{MSE}(\hat{\mathrm{U}}) = \min_{\pi \in \mathrm{Sym}(\mathcal{C})} \frac{1}{n} \mathbb{E} ||\pi(\hat{\mathrm{U}}) - \mathrm{U}_\star||_F \tag{5}$$

where $|| \cdot ||_F$ denotes the matrix Frobenius norm. In particular, we will be interested in characterizing reconstruction in the *proportional high-dimensional limit* where the number of samples $n$, the ambient

dimension $d$ and the sparsity level $s$ go to infinity $n, d, s \to \infty$ at fixed density $\rho = s/d$, sample complexity $\alpha = n/d$, number of clusters $k \geq 2$ and signal strength $\lambda > 0$.

Note that the clustering problem above is closely related to the problem of estimating the class means / features $V_\star$. Indeed, written as in eq. (4) the problem of estimating both the labels and centroids $(U_\star, V_\star)$ boils down to a low-rank matrix factorization problem. In this manuscript, we have chosen to frame the discussion in terms of the clustering, but all our results could be presented also in terms of the reconstruction of the class means.

In this manuscript we provide a sharp asymptotic characterization of when reconstruction, as measured by positive correlation with the ground truth, is possible in high-dimensions for this model, both *statistically* and *algorithmically*. In particular, our **main contributions** are:

- We map the subspace clustering problem to an asymmetric matrix factorization problem. We compute the closed-form asymptotic formula for the performance of the Bayesian-optimal estimator in the high-dimensional limit, building on a statistical physics inspired approach [2] that has been rigorously proven in this case [3, 4] . This allows us to provide a sharp threshold bellow which reconstruction of the features is *statistically* impossible as a function of the parameters of the model.
- To estimate the algorithmic limitations of reconstruction, we analyse a tailored approximate message passing (AMP) algorithm [2, 5, 6] approximating the posterior marginals in this problem, and derive the associated *state evolution equations* characterising its asymptotic performance. Such algorithms are optimal among first order methods [7, 8]) and therefore their reconstruction threshold provides a bound on the algorithmic complexity clustering in our model.
- The two results above allow us to paint a full picture of the *statistical-to-algorithmic* trade-offs in high-dimensional subspace clustering for the sparse $k$-Gaussian mixture model, and in particular to identify an *algorithmically hard* region of the sparsity level $(1 - \rho)$ vs. signal-to-noise ratio $(\lambda)$ plane for which reconstruction is possible statistically but not algorithmically, see Fig. 1. Further, we provide a detailed analysis in the high sparsity regime $(\rho \to 0^+)$ of how the algorithmically hard region grows as we increase the number of clusters and the sparsity level. In particular, the *information theoretical* transition arises at $\lambda_{\mathrm{it}} \approx \sqrt{-k\rho \log \rho}/\sqrt{\alpha}$, and the *algorithmic* one at $\lambda_{\mathrm{alg}} \geq k/\sqrt{\alpha}$.
- The analysis for AMP optimality relies on the finite $\rho$ assumption as $n, d \to \infty$. We thus also investigate the complementary case when the number of non-zero components is of the order $s \lesssim n$ and indeed see that sparse principal component analysis (SPCA) and Diagonal thresholding [9] can perform better than random in this region. We find, however, that this requires $s \leq \sqrt{n}$ (thus $\rho = o(1)$). We rephrase our findings in terms of existing literature on the subspace clustering for two-classes Gaussian mixtures [10, 11].

**Related works:** Subspace clustering is a well-studied topic in classical statistics, with a wide range of methods used in practice, see [12] for a review. Closer to this work is the theoretical line of work studying the limitations of clustering in high-dimensions. Baik, Ben Arous and Péché have shown that PCA for Gaussian mixture clustering fails to correlate with the mixture means below certain threshold known as the *BBP transition* [13]. For dense Gaussian means, the statistical and algorithmic limitations of clustering were analysed in different regimes of interest. Our approach to study subspace clustering relies on a mapping to a low-rank matrix factorization problem. Low-rank matrix factorization has been extensively studied in the literature, and its asymptotic Bayes-optimal performance was characterized in [2–5, 14–16]. The construction of AMP algorithms and the associated state evolutions for matrix factorization was done in [2, 5, 6, 17]. In this work, we leverage these general results on matrix factorization. The closest to our work is perhaps [2, 18], where the authors characterize the asymptotic reconstruction thresholds for the dense case $(\rho = 1)$ in the proportional limit where the number of samples and input dimensions diverge at a constant rate. Non-asymptotic results were also discussed in [19] which considered a modification of Lloyd's algorithm [20] achieving minimax optimal rate and proving a computational lower bound. To the best of our knowledge, the case in which the means are sparse has only been analysed in the regime where the number of non-zero components is sub-extensive with respect to the input dimension. Lower bounds for the statistically optimal recovery threshold in this regime were given in [21, 22], while computational lower bounds are studied in [23, 24]. Clustering of two component Gaussian mixtures has been studied in order to shed light on comparison between statistical and algorithmical tractability in a sparse scenario in [10, 11, 25]. In particular, [10] conjectured and [11] proved algorithmic bounds for this problem. They claim that even below the BBP threshold they can build an algorithm

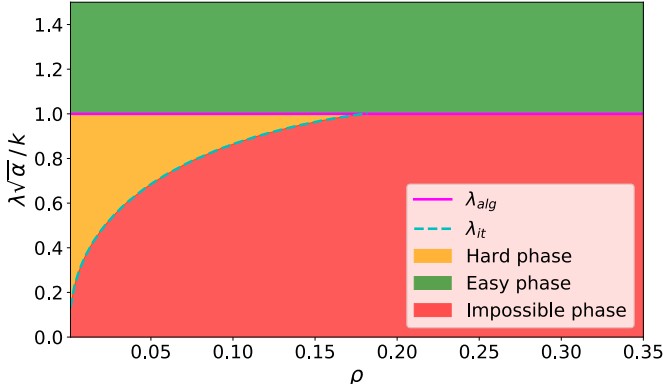

Figure 1: Phase diagram for the subspace clustering of two-clusters GMM at fixed $\alpha = 2$. We plot the SNR $\lambda$ as a function of $\rho$ and we rescale the y-axis by $\sqrt{\alpha}/k$. We colour different region of the figure according to the associated phase. The algorithmic threshold $\lambda_{\text{alg}}$ is the solid line in magenta while the information-theoretic threshold $\lambda_{\text{it}}$ is the dashed line in cyan. In the *impossible* region, no method can perform better than a random guess. In the *hard* region, a partial reconstruction of the signal is theoretically possible, but we do not know of any polynomial time algorithm that can do it. In the *easy* region, however, AMP, can achieve positive correlation with the ground truth (and actually achieves Bayes MMSE, except very close to the tri-critical point, see the discussion in App. A)

achieving exponentially small misclustering error, given that (up-to log factors) we have $s \lesssim \sqrt{n}$. We relate our findings to their work exploring the extreme sparsity regime in detail.

## 3   Main theoretical results

In this section, we introduce the two main technical results allowing us to characterize the limitations of clustering reconstruction (both statistically and algorithmically) for the model introduced above in the proportional high-dimensional limit.

**Statistical reconstruction:**   First, note that up to the permutation symmetry, the estimator minimizing the averaged mean-squared error in eq. (5) admits a closed-form solution given by the marginals of the posterior distribution:

$$\hat{U}_{\text{bo}} = \underset{U \in \mathbb{R}^{n \times k}}{\arg \min} \, \text{MSE}(U) = \mathbb{E}\left[U | X\right] \tag{6}$$

where the posterior distribution for the model defined in eq. (4) explicitly reads:

$$P(U|X) = \frac{1}{Z(X)} \prod_{\nu=1}^{n} P_u(\boldsymbol{u}_\nu) \int_{\mathbb{R}^k} \prod_{i=1}^{d} \left(\mathrm{dv}_i P_v(\boldsymbol{v}_i)\right) \prod_{\nu=1}^{n} \prod_{i=1}^{d} e^{-\frac{1}{2}\left(X_{\nu i} - \sqrt{\frac{\lambda}{s}} \boldsymbol{u}_\nu^\top \boldsymbol{v}_i\right)^2} \tag{7}$$

and for convenience we defined the vectors $\boldsymbol{u}_\nu, \boldsymbol{v}_i \in \mathbb{R}^k$ which are the rows of U, V, and with the prior distribution $P_u$ being the uniform distribution over the indicator vectors defined in eq. (3) and $P_v$ given by the Gauss-Bernouilli distribution defined in eq. (2).

Although in principle it would be possible to compute the minimum mean-squared error (MMSE) given by the Bayes-optimal estimator in eq. (6) by sampling from the posterior from eq. (7), this is impractical when $d, n$ are large. For instance, simply computing the different integrals involved in the evidence $Z$ scale exponentially with the dimensions. The first result consists precisely in a closed-form solution for the asymptotic performance of the Bayes-optimal estimator:

**Main theoretical result 1** *In the proportional high-dimensional limit where $n, d, s \to \infty$ with fixed ratios $\rho = {}^s\!/d$, $\alpha = {}^n\!/d$ and fixed $\lambda, k$, the minimum mean-squared error for the reconstruction of $U \in \mathbb{R}^{n \times k}$ is given by:*

$$\lim_{n \to \infty} MMSE = \frac{k-1}{k} - \text{Tr} \, M_u^\star \tag{8}$$

where $M_u^\star \in \mathbb{R}^{k\times k}$ is the solution of the following minimization problem:

$$M_u^\star = \underset{M_u \in \mathbb{R}^{k\times k}}{\arg\min} \left\{ \max_{M_v \in \mathbb{R}^{k\times k}} \Phi_{rs}\left(M_u, M_v\right) \right\}. \tag{9}$$

with:

$$\Phi_{rs}(M_u, M_v) = \frac{\alpha\lambda\,\mathrm{Tr}\,M_u M_v}{2\rho} - \mathbb{E}_{\boldsymbol{v}_*,\boldsymbol{w}} \left[ \log Z_v \left( \frac{\alpha\lambda M_u}{\rho}, \frac{\alpha\lambda M_u \boldsymbol{u}_*}{\rho} + \sqrt{\frac{\alpha\lambda M_u}{\rho}}\boldsymbol{w} \right) \right]$$
$$- \alpha\,\mathbb{E}_{\boldsymbol{u}_*,\boldsymbol{w}} \left[ \log Z_u \left( \frac{\lambda M_v}{\rho}, \frac{\lambda M_v \boldsymbol{v}_*}{\rho} + \sqrt{\frac{\lambda M_v}{\rho}}\boldsymbol{w} \right) \right] \tag{10}$$

where we introduced $\boldsymbol{u}_* \sim P_u, \boldsymbol{v}_* \sim P_v, \boldsymbol{w} \sim \mathcal{N}(0, I_k)$, and we defined $Z_{u/v}$ as:

$$Z_u(A, \boldsymbol{b}) = \frac{1}{k} \sum_{c\in\mathcal{C}} \exp\left( \boldsymbol{b}^\top \boldsymbol{u}_c - \frac{1}{2}\boldsymbol{u}_c^\top A \boldsymbol{u}_c \right) \tag{11}$$

$$Z_v(A, \boldsymbol{b}) = 1 - \rho + \rho\exp\left( \frac{\boldsymbol{b}^\top (I_k + A)^{-1}\boldsymbol{b}}{2} \right)\sqrt{\det\left(I_k + A\right)^{-1}} \tag{12}$$

Result 1 follows from our mapping of the subspace clustering problem introduced in Sec. 2 to a low-rank matrix factorization form eq. (4). Indeed, this mapping allows us to leverage a closed-form formula characterizing the asymptotic MMSE for low-rank matrix estimation with generic priors that was derived heuristically [2, 18] using the replica method from Statistical Physics and was rigorously proven in a series of works [3–5, 14] in the context of subspace clustering. Our contribution resides in making this connection and drawing the consequences for the subspace clustering problem, a non-trivial endeavour given the complexity of the resulting formulas. Note that the minimization problem eq. (9) is fundamentally different from the one in eq. (6). Indeed, the first involves only low-dimensional variables, and can be easily solved in a computer, while the latter is a high-dimensional problem which is computationally intractable for large $d, n$. The other parameter $M_v^\star \in \mathbb{R}^{k\times k}$ solving eq. (9) can be used to characterize the MMSE reconstruction error on V.

**Algorithmic reconstruction:** While result 1 allow us to sharply characterize when clustering is statistically possible, it does not provide us a practical way to estimate the true class labels $U_\star \in \mathbb{R}^{n\times k}$ from the data $X \in \mathbb{R}^{d\times n}$. In order to provide a bound in the algorithmic complexity clustering, we consider an *approximate message passing* (AMP) algorithm for our problem. Message passing algorithms are a class of first order algorithms (scaling as $O(nd)$, the dimensions of the input matrix X) designed to approximate the marginals of a target posterior distribution, and which have two very important features. First, for a large class of random problems (such as the clustering problem studied here) AMP provides the best known first order method in terms of estimation performance [26–30], and has been rigorously proven to be the optimal for certain problems [7, 8]. Second, the asymptotic performance of AMP can be tracked by a set of low-dimensional *state evolution equations* [31], meaning that its reconstruction performance can be sharply computed without having to run a high-dimensional instance of the problem. For low-rank matrix factorization problems an associated AMP algorithm can be derived [5, 6, 17, 32, 33]. Therefore, yet again we leverage the mapping of subspace clustering to a low-rank matrix factorization problem to derive the associated AMP algorithm 1 with denoising functions $\eta_v, \eta_u$:

$$\eta_u(A, \boldsymbol{b}) = \frac{1}{\sum_{c=1}^k \exp\left( \boldsymbol{b}^\top \boldsymbol{u}_c - \frac{\boldsymbol{u}_c^\top A \boldsymbol{u}_c}{2} \right)} \sum_{c=1}^k \boldsymbol{u}_c \exp\left( \boldsymbol{b}^\top \boldsymbol{u}_c - \frac{\boldsymbol{u}_c^\top A \boldsymbol{u}_c}{2} \right) \tag{13}$$

$$\eta_v(A, \boldsymbol{b}) = \frac{(I_k + A)^{-1}\boldsymbol{b}}{\rho + (1-\rho)\sqrt{\det\left(I_k + A\right)}\exp\left( \frac{-\boldsymbol{b}^\top (I_k+A)^{-1}\boldsymbol{b}}{2} \right)} \tag{14}$$

As mentioned above, one of they key features of Algorithm 1 is that its asymptotic performance can be tracked exactly by a set of low-dimensional equations.

**Algorithm 1** low-rAMP

**Input:** Data $X \in \mathbb{R}^{d \times n}$
Initialize $\hat{\boldsymbol{v}}_i^{t=0}, \hat{\boldsymbol{u}}_\nu^{t=0} \sim \mathcal{N}(\mathbf{0}_k, \epsilon I_k)$, $\hat{\sigma}_{u,\nu}^{t=0} = 0_{k \times k}$, $\hat{\sigma}_{v,i}^{t=0} = 0_{k \times k}$.
**for** $t \leq t_{\max}$ **do**
$$\mathrm{A}_u^t = \tfrac{\lambda}{s} \left( \hat{\mathrm{U}}^t \right)^\top \hat{\mathrm{U}}, \qquad A_v^t = \tfrac{\lambda}{s} \left( \hat{\mathrm{V}}^t \right)^\top \hat{\mathrm{V}}$$
$$\mathrm{B}_v^t = \sqrt{\tfrac{\lambda}{s}} X \hat{\mathrm{U}}^t - \tfrac{\lambda}{s} \sum_{\nu=1}^{n} \sigma_{u,\nu}^t \hat{\mathrm{V}}^{t-1}, \quad \mathrm{B}_u^t = \sqrt{\tfrac{\lambda}{s}} X^\top V - \tfrac{\lambda}{s} \sum_{i=1}^{d} \sigma_{v,i}^t \hat{\mathrm{U}}^{t-1}$$
Take $\{\boldsymbol{b}_{v,i}^t \in \mathbb{R}^k\}_{i=1}^d$, $\{\boldsymbol{b}_{u,\nu}^t \in \mathbb{R}^k\}_{\nu=1}^n$ rows of $\mathrm{B}_v^t, \mathrm{B}_u^t$
$$\hat{\boldsymbol{v}}_i^{t+1} = \eta_v(\mathrm{A}_v^t, \boldsymbol{b}_{v,i}^t), \qquad \hat{\boldsymbol{u}}_\nu^{t+1} = \eta_u(\mathrm{A}_u^t, \boldsymbol{b}_{u,\nu}^t)$$
$$\hat{\sigma}_{v,i}^{t+1} = \partial_{\boldsymbol{b}} \eta_v(\mathrm{A}_v^t, \boldsymbol{b}_{v,i}^t), \qquad \hat{\sigma}_{u,\nu}^{t+1} = \partial_{\boldsymbol{b}} \eta_u(\mathrm{A}_u^t, \boldsymbol{b}_{v,\nu}^t)$$
Here $\hat{\mathrm{U}}^t \in \mathbb{R}^{n \times k}, \hat{\mathrm{V}}^t \in \mathbb{R}^{d \times k}, \mathrm{B}_u^t \in \mathbb{R}^{n \times k}, \mathrm{B}_v^t \in \mathbb{R}^{d \times k}, \mathrm{A}_u^t \in \mathbb{R}^{k \times k}, \mathrm{A}_v^t \in \mathbb{R}^{k \times k}$
**end for**
**Return:** Estimators $\hat{\boldsymbol{v}}_{\mathrm{amp},i}, \hat{\boldsymbol{u}}_{\mathrm{amp},\nu} \in \mathbb{R}^k, \hat{\sigma}_{u,\nu}, \hat{\sigma}_{v,i} \in \mathbb{R}^{k \times k}$

---

**Main theoretical result 2** *In the proportional high-dimensional limit where $n, d, s \to \infty$ with fixed ratios $\rho = {}^s/d$, $\alpha = {}^n/d$ and fixed $\lambda, k$, the correlation between the ground truth $(U_\star, V_\star)$ and the AMP estimators $(\hat{U}_{amp}^t, \hat{V}_{amp}^t)$ at iterate $t$,*

$$M_u^t = \frac{1}{n} U_\star^\top \hat{U}_{amp}^t, \qquad\qquad M_v^t = \frac{1}{n} V_\star^\top \hat{V}_{amp}^t \qquad (15)$$

*satisfy the following* state evolution equations*:*

$$M_u^{t+1} = \mathbb{E}_{\boldsymbol{u}_* \sim P_u, \boldsymbol{\xi} \sim \mathcal{N}(\mathbf{0}_k, I_k)} \left[ \eta_u \left( \frac{\alpha \lambda M_u}{\rho}, \frac{\alpha \lambda M_u \boldsymbol{u}_*}{\rho} + \sqrt{\frac{\alpha \lambda M_u}{\rho}} \boldsymbol{w} \right) \boldsymbol{u}_*^\top \right] \qquad (16)$$

$$M_v^{t+1} = \mathbb{E}_{\boldsymbol{v}_* \sim P_v, \boldsymbol{\xi} \sim \mathcal{N}(\mathbf{0}_k, I_k)} \left[ \eta_v \left( \frac{\lambda M_v}{\rho}, \frac{\lambda M_v \boldsymbol{v}_*}{\rho} + \sqrt{\frac{\lambda M_v}{\rho}} \boldsymbol{w} \right) \boldsymbol{v}_*^\top \right] \qquad (17)$$

*Moreover, the asymptotic performance of is simply given by:*

$$\lim_{n \to \infty} MSE(\hat{U}_{amp}^t) = \frac{k-1}{k} - \mathrm{Tr}\, M_u^t \qquad (18)$$

Result 2 is a consequence from the general theory connecting AMP algorithms to their state evolution (SE) [31]. In the context of low-rank matrix factorization, a derivation of the state evolution equations above from Algorithm 1 were first provided for the rank-one case in [5, 6] and were extended to general rank and denoising functions in [2, 18]. A crucial observation is that Result 2 is intimately related to Result 1. Indeed, the state evolution equations (17) coincide exactly with running gradient descent on the potential defined in eq. (10). While the performance of the statistically optimal estimator $\hat{U}_{\mathrm{bo}}$ is given by the fixed point with the minimal value of the potential $\Phi_{\mathrm{rs}}$, the performance of the AMP estimator $\hat{U}_{\mathrm{amp}}$ is described by the closest minima to the initialization $M_u^{t=0}$. Therefore, studying both the statistical and algorithmic limitations of clustering in high-dimensions boils down to the study of the minima of the potential $\Phi_{\mathrm{rs}}$, see App. A for further details. The identification of a subspace clustering problem with a matrix factorization one allows us to leverage the rich literature from this field. However, analyzing these formulas is far from trivial and consitutes a considerable technical challenge. The major difficulty is to find a suitable parametrization of the overlap matrices that reduces the number of parameters to be tracked, we discuss this in detail in Sec. 5. Moreover, we find the scaling ansatz in the high-sparsity limit that closes the equations on amenable quantities in order to find analytically the statistical-to-computational gap for the large-rank setting, see Sec. 6. However, we stress that we did not take all the necessary precautions to claim full rigor. e.g. prove formally that the minimum is unique (although we checked all these both analytically and numerically).

## 4   Reconstruction limits for sparse clustering

These results allow us to paint a full picture for when sparse subspace clustering is possible in the model defined in Sec. 2 as a function of the quantity of data $\alpha$, the sparsity $1-\rho$, the number of clusters

$k$ and the SNR $\lambda$. Figure 1 summarizes the different reconstruction regimes in the $(\rho, \lambda)$ plane for a two-clusters problem at fixed sample complexity $\alpha = 2$, also known as a *phase diagram*. Moreover the general considerations on the reconstruction limits for sparse clustering, given by analyzing the two-clusters problem, are easily generalizable to the general mixture case and not restricted to that particular model, see Sec. 5. For a fixed sparsity $1 - \rho$, we identify the following regions in Fig. 1:

**Impossible phase:** There is not enough information in the data matrix $X$ handled to the statistician to assign cluster membership better than chance for *any algorithm*. The Bayes-optimal MMSE is not better than a random guess. Clustering (reconstruction of $U_\star$ better than chance) is impossible.

**Hard phase:** The MMSE is non-trivial, and clustering is statistically possible to some extent, but the best known polynomial time algorithm, AMP, fails to correlate better than chance with the true cluster assignment $U_\star$. Any polynomial-time algorithm is conjectured to fail in this region.

**Easy phase:** In the easy phase, not only clustering is statistically possible, but AMP is able to achieve positive correlation with $U_\star$. One can also investigate when AMP achieves the Bayes-optimal MMSE (instead of just positive correlation) leading to the same transition (except from a subtle correction very close to the tri-critical point, see the discussion in App. A).

In Fig. 2, we investigate the different phases presented above by varying the sparsity level $1 - \rho$ and SNR $\lambda$ at fixed $\alpha$. We compare the performance of AMP with popular spectral algorithms: Principal Component Analysis (PCA) and Sparse Principal Component Analysis (SPCA). We can initialize AMP and the SE equations in two different ways: we call *uninformative initialization* a choice for the first iterates of AMP and SE which assumes no knowledge on the ground truth values; conversely with *informative initialization* we consider that the statistician has some prior knowledge on the the ground truth signal. Note that in a real-life scenario the statistician does not have access to the ground truth. Yet, as a theoretical tool the informative initialization provides important information about the algorithmically hard phase, see App A for a discussion. The initialization strategy we considered in the uninformed case for Algorithm 1 is not the only possible choice, there are smarter ways of initialising which can lead to a considerable improvement without explicitly assuming any information about the signal, e.g. spectral initialization [34]. Looking at Fig. 2 we see that SE with uninformed initialization tracks AMP. Morover we note that increasing the sparsity level, i.e. decreasing $\rho$, the problem becomes algorithmically harder. This is reflected in a discontinous jump in the MSE at $\lambda_{\mathrm{alg}}$ which becomes larger. We observe along the same lines a neat advantage by imposing the sparsity constraint in the spectral algorithm (SPCA) with respect to vanilla one (PCA) as the sparsity grows. We discuss the details on the numerical simulations in App. D and the code is available at https://github.com/lucpoisson/SubspaceClustering.

## 5   Stability analysis and algorithmic threshold

In this section we provide a detailed analytical derivation of the threshold $\lambda_{\mathrm{alg}}$ characterizing the algorithmic reconstruction as a function of the number of clusters and the sparsity. First, note due to the permutation symmetry of the clusters the overlap matrices admit the following parametrization:

$$\mathbf{M}_u^t = \frac{m_u^t}{k}\mathbf{I}_k - \frac{m_u^t}{k^2}\mathbf{J}_k \qquad\qquad \mathbf{M}_v^t = m_v^t\mathbf{I}_k - \frac{m_v^t}{k}\mathbf{J}_k \qquad (19)$$

where $\mathbf{J}_k$ is the $k \times k$-matrix with all elements equal to one. This parametrization is preserved under the SE iterations. Therefore, inserting it in the SE equations yield equations for $(m_u^t, m_v^t)$:

$$m_u^{t+1} = f_u^{(k)}\big(\lambda m_v^t/\rho\big) \qquad\qquad m_v^{t+1} = f_v^{(k)}\big(\alpha\lambda m_u^t/\rho\big) \qquad (20)$$

where we introduced the following update functions:

$$f_u^{(k)}(z) := \frac{k}{k-1}\,\mathbb{E}_{\boldsymbol{\omega}\sim\mathcal{N}(\mathbf{0}_k,\mathbf{I}_k)}\left[\frac{e^{z+w_1\sqrt{z}}}{e^{\lambda m_v^t+w_1\sqrt{z}}+\sum_{l=2}^k e^{w_l\sqrt{z}}}\right] - \frac{1}{k-1} \qquad (21)$$

$$f_v^{(k)}(z) := \frac{\rho^2 z}{k(k+z)}\int_0^{+\infty}\frac{S_{k-1}}{(2\pi)^{\frac{k}{2}}}\frac{\xi^{k+1}e^{-\xi^2/2}}{\rho+(1-\rho)\big(\frac{k+z}{k}\big)^{\frac{k}{2}}e^{-\frac{\xi^2 z}{2k}}}\,\mathrm{d}\xi \qquad (22)$$

where $S_{k-1}$ is the surface of the $k$-dimensional unitary hypersphere. Recall that $(m_u^t, m_v^t) \in [0,1]^2$ fully characterize the reconstruction performance, with $(m_u^\infty, m_v^\infty) = (0,0)$ corresponding to the performance of a random guess. Conversely, $(m_u^\infty, m_v^\infty) = (1, \rho)$ corresponds to perfect reconstruction

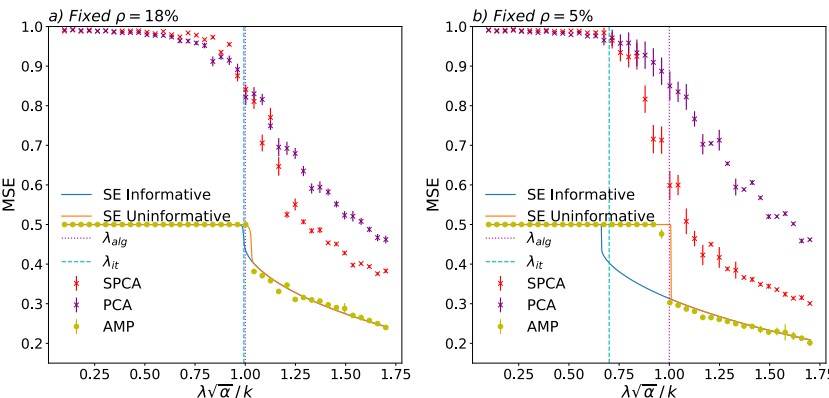

Figure 2: We compare the performance for clustering of two-classes GMM, as measured by the MSE of AMP, SPCA, PCA and SE informed and uninformed. We plot the MSE as a function of the SNR $\lambda$ and we rescale the x-axis by $\sqrt{\alpha}/k$. For each algorithm considered the error bars are built using the standard deviation over fifty runs with parameters ($n = 8000, d = 4000$), i.e. $\alpha = 2$. We plot in vertical line the theoretical values for the Information-Theoretic threshold $\lambda_{it}$ (dashed cyan line), and the algorithmic threshold $\lambda_{alg}$ (dotted line in magenta). The theoretical values coincide with the experimental one. The SE with uninformed initialization follows AMP as expected. *Left*: The sparsity is fixed with parameter $\rho = 18\%$. Both SPCA and PCA have a worse performance with respect to AMP and in this sparsity regime we have only a marginal advantage by using SPCA with respect to PCA. *Right*: The sparsity is fixed with parameter $\rho = 5\%$. Increasing the sparsity level the width of the algorithmically hard phase becomes bigger and SPCA performs clearly better than PCA.

of the cluster membership and the sparse cluster means. One can check that the *trivial fixed point* $(m_u, m_v) = (0, 0)$ is always a fixed point of the SE equations. Moreover, note that its stability is crucially connected to the algorithmic threshold $\lambda_{alg}$. Indeed, if the trivial fixed point $(m_u, m_v) = (0, 0)$ is stable, AMP with an uninformed initialization will always converge to this point - and therefore achieve random guessing performance. To study its stability, we expand the update functions around $(m_u, m_v) = (0, 0)$ up to the second order. Expressing everything in terms of $m_u$, we obtain:

$$m_u^{t+1} = F_{se}\left(m_u^t\right) = \frac{\lambda^2 \alpha}{k^2} m_u^t + \left(\frac{(k-4)\alpha^2 \lambda^4}{2k^4} - \frac{\lambda^3 \alpha^2}{k^3}\right)\left(m_u^t\right)^2 + o\left((m_u^t)^2\right) \qquad (23)$$

This immediately tells us that the trivial fixed point becomes unstable at the *algorithmic threshold* $\lambda_{alg} = k/\sqrt{\alpha}$. This transition is a well-known result in random matrix theory and goes under the name of BBP transition [13]. Despite the fact that we expand around the trivial fixed point, the perturbative method also provides information also about the Bayes-optimal performance, thanks to the general properties that the phase diagrams in Bayes-optimal inference problems must respect [35]. Indeed, a sufficient criterion for the presence of an algorithmically hard phase requires at the algorithmic threshold: $F_{se}''(0) > 0$. The local study predicts that the phase diagram will present an algorithmically hard region for $k > k_{hard} = 4 + 2\sqrt{\alpha}$. The criterion nevertheless is not necessary in this setting, as we immediately see from the phase diagram for the two-components GMM in Fig. 1: we clearly observe an algorithmically hard region for high-sparsity although $k < k_{hard}$. In fact, the *information-theoretic threshold* $\lambda_{it}$, defined as the value of the SNR at which the problem becomes statistically possible, cannot be found simply thanks to the expansion around the trivial fixed point. Indeed the exact computation of $\lambda_{it}$ requires evaluating the potential function $\Phi$ in eq. (10): one needs to find the threshold value of the SNR at which the informative fixed point has a lower value of the free energy with respect to the uninformative fixed point. The technical discussion on the calculation of $\lambda_{it}$, which we see plotted in Fig. 1 for the two-classes case, is given in App. B, while a general overview on the different thresholds is given in App. A. It is interesting to stress that the linearization of the SE equation around the trivial fixed point in eq. (23) coincide to study the gradient of the potential $\Phi_{rs}$ in eq. (10) around that point, and hence the stability of the trivial fixed point is determined by the potential landscape. This connection is at the roots of the conjectured link between the hardness of an inference task and AMP: the statistical-to-computational gap opens up as the fixed point with minimal value of the potential in eq. (10), describing the performance of the Bayes-optimal estimator, is attained far from a locally stable region around the uninformative fixed point, where the AMP algorithm gets stuck.

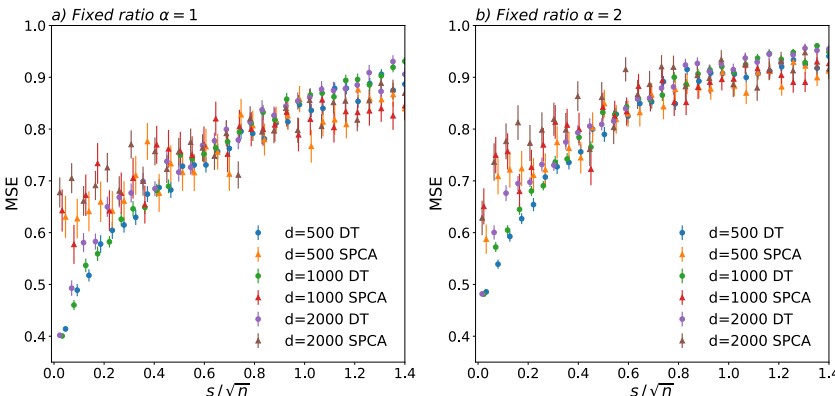

Figure 3: We compare the performance of diagonal thresholding (DT) and Sparse PCA (SPCA), as measured by the MSE, for clustering of two-classes GMM for two different parameter $\alpha$. We plot the MSE vs the number of non-zero component $s$ and we rescale the x-axis by the square-root of the number of samples. The left plot is done at $\alpha = 1$ while the right one is for $\alpha = 2$. The SNR is tuned such that we are always under the BBP threshold. With this choice we always work in the sub-extensive sparsity regime $\rho = o(1)$ and we can verify numerically what has been claimed in the literature [10, 11]: efficient algorithms in the high dimensional limit need a number of non-zero component (up-to log factors) $s \lesssim \sqrt{n}$ in order to beat random guessing under the BBP threshold.

## 6 Large sparsity regime

We characterize in this section the scaling of the thresholds in the very sparse (and most interesting) regime when $\rho \to 0^+$. We highlight the main passages of the computation and focus on the principal results, see Appendix C for a detailed analysis. The starting point is the following change of variables:

$$m_u = \tilde{m}_u \sqrt{\frac{-\rho \log \rho}{\alpha}} \qquad m_v = \tilde{m}_v \rho \qquad \lambda = C(k)k\sqrt{\frac{-\rho \log \rho}{\alpha}} \qquad (24)$$

Inserting these expressions into eq. (20), we obtain simplified SE equations for the rescaled overlaps $(\tilde{m}_u, \tilde{m}_v)$ without any residual dependence on $(\rho, \alpha)$:

$$\tilde{m}_u = C(k)\tilde{m}_v \qquad\qquad \tilde{m}_v = T_k\left(C(k)\tilde{m}_u\right) \qquad (25)$$

where we introduced the auxiliary function $T_k(\cdot)$ defined as:

$$T_k(z) = \int_0^{+\infty} \frac{S_{k-1}}{k(2\pi)^{k/2}} \xi^{k+1} e^{-\xi^2/2} \Theta\left(\frac{z\xi^2}{2} - 1\right) d\xi$$

where $\Theta(\cdot)$ is the Heaviside theta function. By considering the large $k$ expansion of $T_k(\cdot)$, we can derive the scaling of the IT threshold with $(k, \rho, \alpha)$, see Appendix C for more details. Putting together with our previous result for $\lambda_{\text{alg}}$ from Sec. 5, we obtain the following fundamental result:

$$\lambda_{\text{it}} \approx \sqrt{\frac{-k\rho \log \rho}{\alpha}} \qquad\qquad \lambda_{\text{alg}} = \frac{k}{\sqrt{\alpha}}. \qquad (26)$$

These equations completely characterize the behavior at large rank & small (but finite) sparsity. The statistical-to-computational gap, where AMP is not able to exploit the information on the sparse nature of the cluster means, grows with both the sparsity and the rank. In App. C, we further show that the large rank expression for the IT threshold $\lambda_{\text{it}}$ is accurate already at moderate $k \approx 10$, see Fig. 9. It is interesting at this point to reframe our results on the high sparsity regime in the context of the existing literature. The results for the IT transition for the detection of two-classes sparse GMM were discussed in [22, 23], and we obtain a consistent scaling in the small $\rho$ behaviour. Despite this fact, the algorithmic bounds of different relevant works for the same problem [10, 11, 23] seem, at first sight, to not agree with our findings. In particular [11] proves rigorously the existence of an algorithm that achieves minimax rate under the BBP threshold for clustering of two-classes sparse GMM. This apparent inconsistency is related to different sparsity regimes analyzed. Indeed, here we investigate

the extensive sparsity regime, i.e. $\rho = O(1)$, while the guarantees for the efficient algorithms working under the BBP threshold require a very high sparsity level, i.e. $\rho = o(1)$. This difference is crucial. In Fig. 3 we illustrate it by considering the performance of two popular algorithms for this problem: *Diagonal Thresholding* (DT) [9] and SPCA. For extremely large sparsity, i.e. $s = O(1)$, these algorithms indeed provide estimators with positive correlation with the true classes below the BBP threshold! However, as soon as we increase the density of non-zero components the performance strongly deteriorates. In fact, the transition to random chance performance takes place as the number of non-zero components approach $s \lesssim \sqrt{n}$, in agreement with the literature [10, 11].

## 7 Conclusion

In this work we considered the problem of clustering $k$ homogeneous Gaussian mixtures with sparse means. Mapping this to a low-rank matrix factorization problem, we have provided an exact asymptotic characterization of the MMSE in the high dimensional limit. The Bayes-optimal performance was compared to AMP, the best known polynomial time algorithm for this problem in the studied regime. In the large sparsity regime, we uncovered a large statistical-to-computational gap as the sparsity level grows, and unveiled the existence of a computationally *hard phase*. In particular, we have shown that the SNR threshold below which recovery is statistically impossible is given by $\lambda_{\text{it}} \approx \sqrt{-k\rho \log \rho}/\sqrt{\alpha}$, while the one for which AMP positively correlates with the ground truth classes is given by $\lambda_{\text{alg}} \geq k/\sqrt{\alpha}$. Our result for the existence of an algorithmically hard region was compared with the existing literature for this problem, solving an apparent contradiction due to the scaling assumption of the sparsity level with the dimension of the features. We corroborated our findings with the help of algorithms for subspace clustering such as sparse principal component analysis and diagonal thresholding. The mapping of the subspace clustering problem to a low-rank matrix factorization one is flexible and extending this work to more general scenarios is definitely interesting. The main limitation of the current setting is the Gaussian assumption for the noise. One natural idea to go beyond this limitation is to consider other flavours of message passing algorithms, such as vector AMP (VAMP) [36] to study more general rotationally invariant noise distributions. A further prospective direction is to consider inhomegeneous mixture models, i.e. $p_c \neq 1/k$ in eq. (1), in order to study the effect of unbalancedness on the statistical-to-computational gap.

## Acknowledgment

We thank Maria Refinetti for discussions. This work started as a part of the doctoral course *Statistical Physics For Optimization and Learning* taught at EPFL in spring 2021. We acknowledge funding from the ERC under the European Union's Horizon 2020 Research and Innovation Program Grant Agreement 714608-SMiLe, and by the Swiss National Science Foundation grant SNFS OperaGOST, 200021_200390.

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
