

Figure 4: Enriched phase diagram for the subspace clustering of two-clusters GMM at fixed $\alpha = 2$. We plot the SNR $\lambda$ as a function of $\rho$ and we rescale the y-axis by $\sqrt{\alpha}/k$. We colour different region of the figure according to the associated phase. We introduce in the black dotted line the dynamical spinodal threshold $\lambda_{\text{dyn}}$, in the yellow dashdotted line the Bayes-algorithmical threshold $\lambda_{\text{alg-Bayes}}$ and in purple dashed line the jump-Bayes one $\lambda_{\text{jump-Bayes}}$. It is not visible, due to the choice of the axis, the *easy* region, in which AMP performs better than random but not Bayes-optimally. We analyze this in Fig. 6.

## A Analysis of the thresholds

We identified in Sec. 4 different *reconstruction phases* for the subspace clustering problem, characterizing completely the Bayes-optimal and algorithmical performances. We detail in this section the definition of the reconstruction phases and the consequences for the computational and statistical limits of subspace clustering. We discuss in particular an interesting link between the thresholds separating these phases and the potential function $\Phi_{\text{rs}}$ in eq. (10). First, we enlarge the picture on the reconstruction phases we offered in Sec. 3. Along with the impossible, hard and easy phases we can define a further region. We call the *Alg-Bayes* phase, the region of parameters in which the performance of AMP is, not only achieving positive correlation with the ground truth, but achieves the Bayes-optimal performance. We summarize now the complete description:

**Impossible phase:** There is not enough information in the data matrix $X$ handled to the statistician in order to assign cluster membership better than chance for *any algorithm*, and the Bayes-optimal MMSE is not better than a random guess. Clustering (i.e. reconstruction of $U_\star$ better than chance) is impossible.

**Hard phase:** The MMSE is non-trivial, and clustering is statistically possible to some extent, but the best known polynomial time algorithm, AMP, fails to correlate better than chance with the true cluster assignment $U_\star$. Any polynomial-time algorithm is conjectured to fail in this region.

**Easy phase:** In the easy phase, not only clustering is statistically possible, but AMP is able to achieve positive correlation with $U_\star$.

**Alg-Bayes phase:** In the alg-Bayes phase AMP is able to achieve Bayes-optimal positive correlation with $U_\star$.

Following the introduction of the alg-Bayes phase, we find an enriched version of the phase diagram for the two-classes subspace clustering at fixed $\alpha$, see Fig. 4. When we cross from one region to an other we have a *phase transition*. Each phase transition is characterized by different *thresholds*: values of the parameters which signal the onset of a new phase. In Fig. 4 we see different thresholds which were not present in the previous plot in Fig. 1: $\{\lambda_{\text{dyn}}, \lambda_{\text{alg-Bayes}}, \lambda_{\text{jump-Bayes}}\}$. In order to define these quantities, it is useful to highlight the relationship between the thresholds and the minima of the "free energy" $\Phi_{\text{rs}}$ in eq. (10). Let us fix a sparsity level $1 - \rho$, say $\rho = 0.05$, and move vertically on the y-axis starting from the bottom on the y-axis, i.e. $\lambda\sqrt{\alpha}/k \ll 1$. As we increase the value of the SNR we can identify the following thresholds:

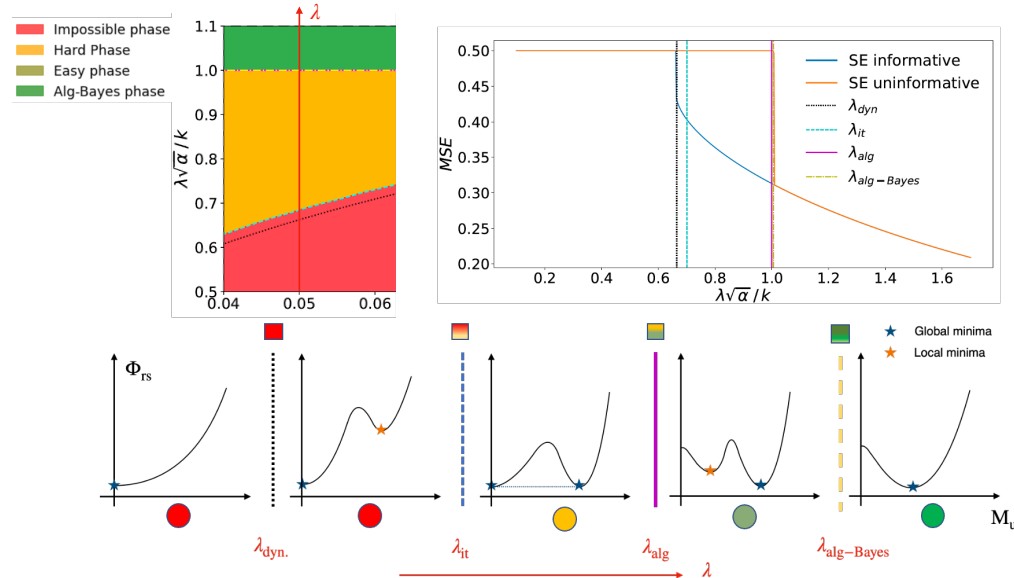

Figure 5: Evolution of the MSE and the potential $\Phi_{rs}$ for fixed $\rho \simeq 0.05$ and $\alpha = 2$. Top: We take vertical cross section of the phase diagram in Fig. 4 for $\rho \simeq 0.05$, as explained in the left panel. In the right panel we analyze the MSE via SE both informed and uninformed as a function of the SNR $\lambda$ and we rescale the x-axis by $\sqrt{\alpha}/k$. In vertical line we plot the different thresholds we encounter as we increase the SNR. Bottom: Cartoon plot of the minima of the potential $\Phi_{rs}$ as we follow the cross section of the phase diagram at $\rho \simeq 0.05$. The blue star denotes the global minimum of $\Phi_{rs}$, which corresponds to the correlation of the Bayesian-optimal estimator, while the orange dot denote local minima. We label the phase in which we are at a given stage by colouring the circle below every plot. We plot the thresholds as vertical lines separating the different subplots.

• $\lambda < \lambda_{\mathrm{dyn}}$: The only minimum of eq. (9) is the trivial minimum corresponding to zero correlation $\mathrm{M}_u = 0$ with $\mathrm{U}_\star$. Therefore, below this threshold reconstruction is impossible.

• $\lambda \in (\lambda_{\mathrm{dyn}}, \lambda_{\mathrm{it}})$: A second minima with higher $\Phi_{rs}$ (i.e. a local minimum) and correlation appears, but the trivial minimum $\mathrm{M}_u = 0$ is still the global one. Therefore, AMP with a uninformed initialization $\mathrm{M}^{t=0} = 0$ will converge to the trivial minimum, and reconstruction is only possible with a strong informed initialization. We call the threshold value for the emergence of this local minimum the *dynamical spinodal* $\lambda_{\mathrm{dyn}}$.

• $\lambda \in (\lambda_{\mathrm{it}}, \lambda_{\mathrm{alg}})$: As the SNR is increased, the local minumum goes down in energy $\Phi_{rs}$, and at a certain $\lambda_{\mathrm{it}}$, it crosses the trivial minimum. Therefore, in this interval the non-trivial minimum is the global one, while the trivial minimum becomes local. Although reconstruction is statistically possible in this region, AMP with a uninformed initialization $\mathrm{M}^{t=0} = 0$ is stuck at the trivial minima. Therefore, in this region we enter the hard phase.

• $\lambda \in (\lambda_{\mathrm{alg}}, \lambda_{\mathrm{alg\text{-}Bayes}})$ As the SNR is further increased, AMP with a uninformed initialization starts to achieve positive correlation with $\mathrm{U}_\star$, although strictly lower than the Bayes-optimal estimator. In terms of the potential $\Phi_{rs}$, this corresponds to the trivial minimum continuously becoming a local maximum, and another local minimum corresponding to higher correlation continuously appearing. This new local minima coexists with the global one, which corresponds to the Bayes-optimal performance. We enter the easy phase.

• $\lambda > \lambda_{\mathrm{alg\text{-}Bayes}}$: Finally, as the SNR is further increased the local minima disappears, and there is only one minimum with high-correlation with the signal left. In this region, AMP with a uninformed initialization achieves the same performance as the Bayes-optimal estimator. We enter the alg-Bayes phase.

The discussion above is summarized in Fig. 7. We plot togheter with the evolution of the minima of $\Phi_{rs}$, the performance of SE with both informative and uninformative initialization to analyze the

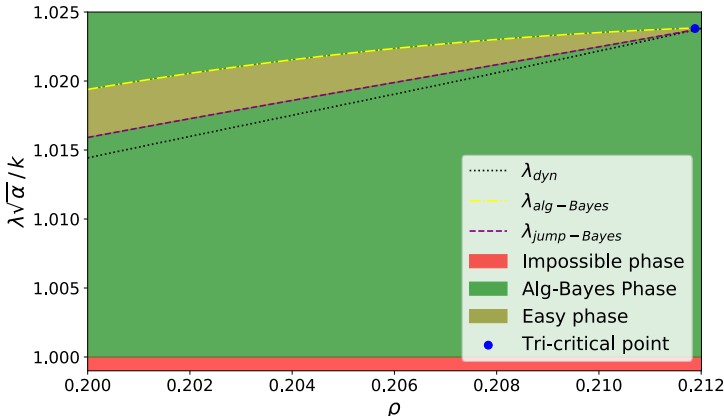

Figure 6: Zoom around the tri-critical point of Fig. 4.

behaviour of the MSE. The full characterization of the subspace clustering problem both from an algorithmic and statistical perspective, boils down to the analysis of the evolution of the critical points of $\Phi_{\text{rs}}$ as we vary the meaningful parameters in the problem. We note from Fig. 4 that the performance of AMP is, when it achieves positive correlation with the ground truth, almost everywhere Bayes-optimal apart from a small region around the *tri-critical* point. This point is defined - at fixed $(\alpha, k)$- as the tuple of parameters $(\lambda_T(\alpha, k), \rho_T(\alpha, k))$, such that the "spinodal" thresholds meet, i.e. $\lambda_{\text{alg-Bayes}} = \lambda_{\text{dyn}}$. We discuss why these thresholds are called *spinodals*, and how to compute them practically in Sec. B. We can analyze the vicinity of the tri-critical point to analyze the non-trivial interplay between the easy and alg-Bayes phase, where AMP does not achieve always Bayes-optimal performance. Imagine to repeat the same steps as before considering the zoom around the tri-critical point of the phase diagram, see Fig. 6. Fix a sparsity level $1 - \rho$, say $\rho = 0.202$, and move vertically on the y-axis starting from the bottom on the y-axis, i.e. $\lambda\sqrt{\alpha}/k \ll 1$. As we increase the SNR we can repeat the previous analysis, obtaining now:

• $\lambda < \lambda_{\text{alg}}$: The only minimum of eq. (9) is the trivial minimum corresponding to zero correlation $M_u = 0$ with $U_\star$. Therefore, below this threshold reconstruction is impossible.

• $\lambda \in (\lambda_{\text{alg}}, \lambda_{\text{dyn}})$: The trivial minimum becomes unstable and AMP achieves positive correlation with the ground truth. The minimum is unique and also SE with a positive initialization would end up there. The phase is alg-Bayes.

• $\lambda \in (\lambda_{\text{dyn}}, \lambda_{\text{jump-bayes}})$: As the SNR is increased, a new local minimum appears. The reconstruction phase is still alg-Bayes since the non-trivial minimum has higher free energy than the global one.

• $\lambda \in (\lambda_{\text{jump-Bayes}}, \lambda_{\text{alg-Bayes}})$ As the SNR is further increased, the free energy of the informative minimum goes down and becomes equal to the uninformative one. We enter the easy phase, nevertheless AMP achieves positive correlation with the truth, Bayes optimal performance is superior. The Bayes-optimal MSE have a first order phase transition at $\lambda_{\text{jump-Bayes}}$, hence the name *jump-Bayes*.

• $\lambda > \lambda_{\text{alg-Bayes}}$: Finally, as the SNR is further increased the local minima disappears, and there is only one minimum with high-correlation with the signal left. In this region, AMP with a uninformed initialization achieves the same performance as the Bayes-optimal estimator. We enter the alg-Bayes phase.

The analysis of the evolution of the minima of $\Phi_{\text{rs}}$ and the consequences on the MSE is done in Fig. 7.

## B Building the phase diagram

We build in this section the phase diagram for the two-classes GMM shown in Fig. 1 , explaining the steps which are easily generalizable to the general mixture case. First we note that we can simplify the model for two-mixtures GMM in eq. (4) even further by mapping it to an easier rank $k = 1$

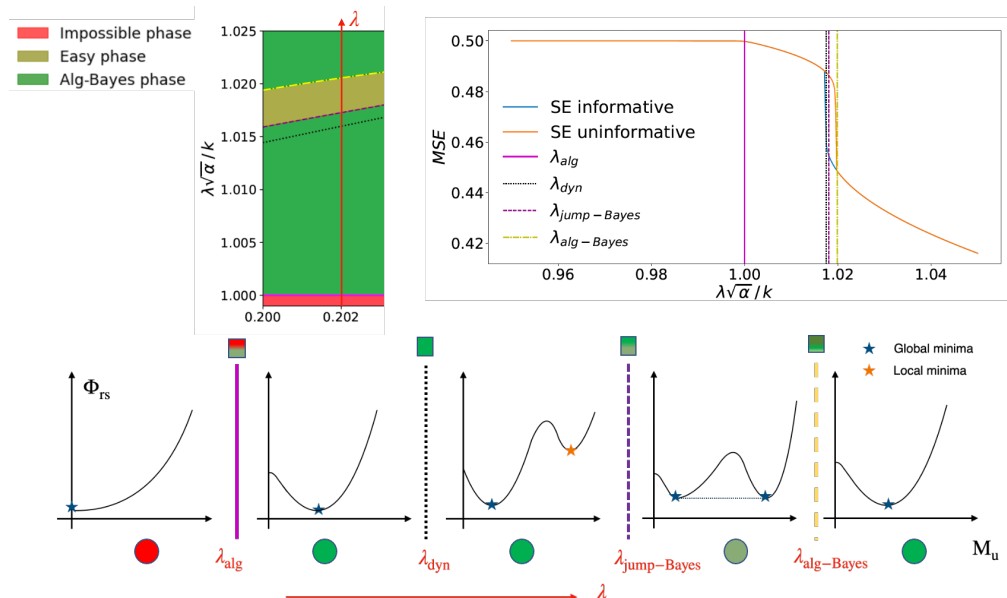

Figure 7: Evolution of the MSE and the potential $\Phi_{\mathrm{rs}}$ for fixed $\rho \simeq 0.202$ and $\alpha = 2$. Top: We take vertical cross section of the phase diagram in Fig. 4 for $\rho \simeq 0.202$, as explained in the left panel. In the right panel we analyze the MSE via SE both informed and uninformed as a function of the SNR $\lambda$ and we rescale the x-axis by $\sqrt{\alpha}/k$. In vertical line we plot the different thresholds we encounter as we increase the SNR. Bottom: Cartoon plot of the minima of the potential $\Phi_{\mathrm{rs}}$ as we follow the cross section of the phase diagram at $\rho \simeq 0.202$. The blue star denotes the global minimum of $\Phi_{\mathrm{rs}}$, which corresponds to the correlation of the Bayesian-optimal estimator, while the orange dot denote local minima. We label the phase in which we are at a given stage by colouring the circle below every plot. We plot the thresholds as vertical lines separating the different subplots.

version of the matrix factorization problem. It suffices to replace the matrices $(U, V)$ in eq. (4) by the following quantities:

$$\boldsymbol{u} \sim \mathrm{Rad}(n) \in \{-1, +1\}^n \qquad \boldsymbol{v} \sim_{i.i.d.} \rho \mathcal{N}(0, \mathrm{I}_d) + (1 - \rho)\delta_0 \in \mathbb{R}^d \qquad (27)$$

The two formulations of the problem are indeed formally equivalent up to a rescaling of the parameters such that at fixed $\alpha$ the quantity $\lambda/k$ is the same in two settings. The mapping simplify significantly the computation. First, in order to compute the Bayes-optimal performance in eq. (8), we must compute the *partition functions* $Z_{u/v}$ in eqs. (11),(12) for the new simplified model. We shall exploit the following general relation, as a function of the prior distribution on $(\mathrm{U}, \mathrm{V})$:

$$Z_{u/v}(\mathrm{A}, \boldsymbol{b}) = \mathbb{E}_{\boldsymbol{x} \sim P_{u/v}}\left[\exp\left(-\boldsymbol{b}^\top \boldsymbol{x} + \frac{\boldsymbol{x}^\top \mathrm{A} \boldsymbol{x}}{2}\right)\right] \qquad (28)$$

thus exploiting the explicit expression of the prior in eq. (27) we obtain:

$$Z_u(A, b) = e^{-\frac{A}{2}} \cosh b \qquad (29)$$

$$Z_v(A, b) = 1 - \rho + \frac{\rho}{\sqrt{1 + A}} \exp\left(\frac{b^2}{2(1 + A)}\right) \qquad (30)$$

We study the computational limits for the subspace clustering problem deriving the associated AMP to this simplified low-rank matrix factorization problem. We have to compute the *denoising functions* for the simplified model, written in eqs. (13),(14) for the general rank case. We shall exploit the general formula relating them with the partition functions computed above:

$$\boldsymbol{\eta}_{u/v}(\mathrm{A}, \boldsymbol{b}) = \partial_{\boldsymbol{b}} \log Z_{u/v}(\mathrm{A}, \boldsymbol{b}) \qquad (31)$$

thus using the prior for the simplified $k = 1$ model in eq. (27) we obtain:

$$\eta_u(A, b) = \tanh(b) \tag{32}$$

$$\eta_v(A, b) = \frac{\rho b}{1 + A} \frac{1}{\rho + (1 - \rho)\sqrt{1 + A}e^{-\frac{b^2}{2(1+A)}}} \tag{33}$$

where now $(A, b) \in \mathbb{R}^2$. We can write at this point the SE equations for the overlaps $(m_u^t, m_v^t)$ which are now scalar variables. Let us introduce the following notation:

$$m_u^{t+1} = \mathbb{E}_{u_* \sim P_u, \xi \sim \mathcal{N}(0,1)} \left[ \eta_u \left( \frac{\lambda m_v^t}{\rho}, \frac{\lambda m_v^t u_*}{\rho} + \sqrt{\frac{\lambda m_v^t}{\rho}} \xi \right) u_* \right] := \mathcal{U}(\lambda m_v^t / \rho) \tag{34}$$

$$m_v^{t+1} = \mathbb{E}_{v_* \sim P_v, \xi \sim \mathcal{N}(0,1)} \left[ \eta_v \left( \frac{\alpha \lambda m_u^t}{\rho}, \frac{\alpha \lambda m_u^t v_*}{\rho} + \sqrt{\frac{\alpha \lambda m_u^t}{\rho}} \xi \right) v_* \right] := \mathcal{V}(\alpha \lambda m_u^t / \rho) \tag{35}$$

The perturbative method we presented in Sec 5 would pinpoint easily the expected algorithmical threshold $\lambda_{\text{alg}} = 1/\sqrt{\alpha}$. Despite this, as we discussed in Sec. 5, it would not guarantee the presence of an algorithmically hard region since the number of cluster $k < k_{\text{hard}}$. We have to resort to numerical computations for finding the exact values of the thresholds since also in this simple case we do not have a closed-form update for the iterates $(m_u^t, m_v^t)$, although we will treat them analytically in App. C. We keep in mind the picture in Figs. (5,7) and compute the different thresholds there defined. Let us consider first $\lambda_{\text{it}}$, the IT threshold, defined as the SNR at which the problem becomes statistically possible. We see in Fig. 5 that it coincides with the SNR level at which the free energy of the two minima (if present) are equal. We analyze for simplicity sparsity levels in which $\lambda_{\text{dyn}} < \lambda_{\text{alg}}$, i.e. we refer to Fig. 5, otherwise the criterion above would define equivalently $\lambda_{\text{bayes-Jump}}$. We compute the difference of free energy $\Delta\Phi$ between the two minima introducing the path $\gamma : \mathbb{R} \to \mathbb{R}^2$ which follows the state evolution equations:

$$\gamma(t) = (t, \mathcal{V}(\alpha \lambda t)) \tag{36}$$

We can use the fundamental theorem of calculus to obtain the difference of free energy between the trivial fixed point $(m_u, m_v) = (0, 0)$ and a non-trivial one $(m_u, m_v) = \big(x, \mathcal{V}(\alpha \lambda(x)x)\big)$ at overlap $m_u = x$ as follows:

$$\Delta\Phi(x) = \int_0^x \mathrm{d}q \frac{d\Phi}{dt}\big(m_u(t) = q, m_v(t) = \mathcal{V}(\alpha \lambda(x)q); \lambda(x), \rho, \alpha\big) \tag{37}$$

where we introduced $\lambda(x)$ as the SNR needed in order to have at fixed $(\rho, \alpha)$ an overlap $m_u = x$ defined by the self-consistent equation:

$$x = \mathcal{U}\big(\lambda(x)\mathcal{V}(\alpha \lambda(x)x)\big) \tag{38}$$

Plugging in the expression of the derivative we identify the IT threshold $\lambda_{\text{it}}$ as the minimal SNR such that the following equation is satisfied:

$$\int_0^{x(\lambda_{it})} dq \mathcal{V}'(\alpha \lambda_{it} q)[q - \mathcal{U}(\lambda_{it}\mathcal{V}(\alpha \lambda_{it} q))] = 0 \tag{39}$$

Let us consider now the Bayes-algorithmical and dynamical thresholds, always referring to their pictorial representation in Figs. (5,7). From a practical standpoint they are stationary point of the the function $\lambda(m_u)$, solution of eq. (38). A reader with some statistical physics background may recognize a parallel with the theory of real gases. The curve $\lambda(m_u)$, exactly as the Pressure-Volume curve $p(v)$ for real gases, is composed of two branches called *stable* and *unstable* branch defined from the value of the derivative $\partial\lambda/\partial m$ (resp. $\partial p/\partial v$). The operative definition of these thresholds as critical points of the curve $\lambda(m_u)$ allows us to easily compute them numerically, see Fig. 8 to observe their evolution as a function of the sparsity level. We see that as we increase the sparsity, i.e. decrease $\rho$, the statistical-to-computation gap, measured visually by the distance between the two critical points, increase. Moreover the IT threshold collapse with the dynamical one. The same happens with the Bayes-algorithmic threshold, approaching $\lambda_{\text{alg}} \approx 1$.

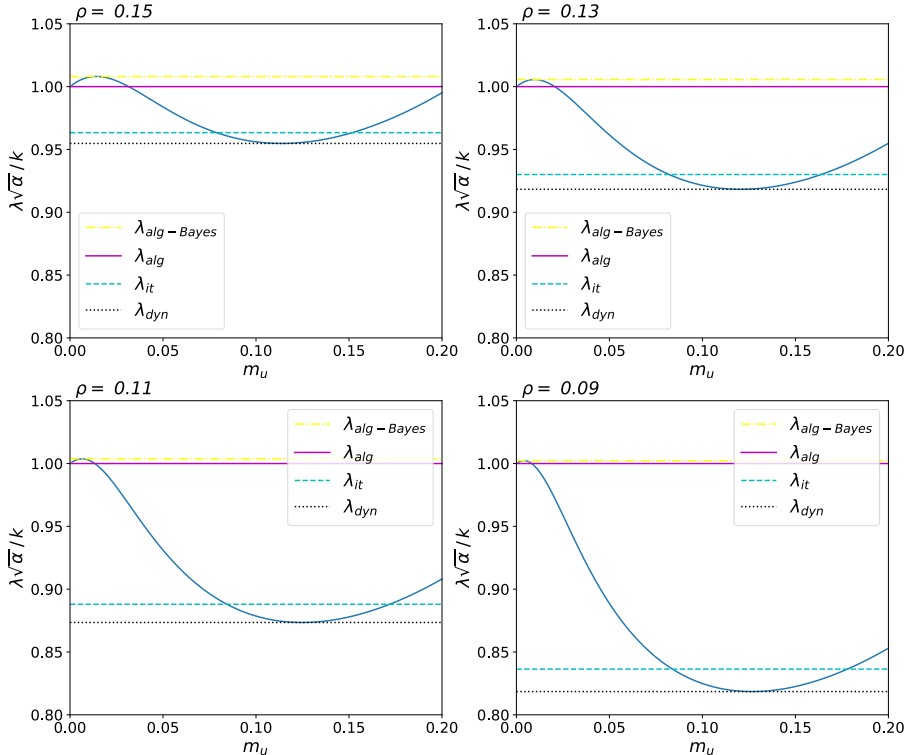

Figure 8: Evolution of the SNR $\lambda$ as a function of $m_u$ for different $\rho \in \{0.09, 0.11, 0.13, 0.15\}$. We rescale the y-axis by $\sqrt{\alpha}/k$. We plot the different transitions $\{\lambda_{\text{dyn}}, \lambda_{\text{it}}, \lambda_{\text{alg}}, \lambda_{\text{alg-Bayes}}\}$ for each different sparsity level. It is visually clear that the hard phase becomes bigger and bigger as the sparsity grows, while the gap between dynamical and IT threshold closes.

## C   Scaling behaviour at large sparsity

We do not have in general an analytical expression for the iterates of the SE equation $(m_u^t, m_v^t)$ appearing in eq. (20), thus we introduced in Sec. 6 a change of variables that allows us to approach analytically the problem in the large sparsity regime. We study in this section the consequences of this scaling assumption. Let us rewrite it here:

$$m_u = \tilde{m}_u \sqrt{\frac{-\rho \log \rho}{\alpha}} \qquad m_v = \tilde{m}_v \rho \qquad \lambda = C(k) k \sqrt{\frac{-\rho \log \rho}{\alpha}} \qquad (40)$$

Consider the update of the parameter $m_v$ under this parametrization:

$$m_v = \rho \tilde{m}_v = f_v^{(k)}\big( - \tilde{m}_u C(k) \log \rho \big) \qquad (41)$$

We can rewrite the right hand side in the following way:

$$\rho \tilde{m}_v = \rho \frac{-\tilde{m}_u C(k) \log \rho}{(k - \tilde{m}_u C(k) \log \rho)} \int_0^{+\infty} \frac{S_{k-1}}{(2\pi)^{\frac{k}{2}}} \frac{\rho \xi^{k+1} e^{-\xi^2/2}}{\rho + (1-\rho)(\frac{k - \tilde{m}_u C(k) \log \rho}{k})^{\frac{k}{2}} \rho^{\tilde{m}_u C(k) \xi^2/2}} \, d\xi \qquad (42)$$

where $S_{k-1}(1)$ is the surface of the $k-$dimensional unitary hypersphere. Working in the small $\rho$ limit allows us to exploit a concentration in measure over which we integrate on the right hand side. The exponent of $\rho$ in the denominator, i.e. $\frac{\tilde{m}_u C(k) \xi^2}{2}$, will determine the large sparsity behaviour. If

the exponent is greater than one, $\rho^{\tilde{m}_u C(k) \xi^2/2}$ will go to zero, otherwise it will diverge in the limit $\rho \to 0$. Thus we obtain:

$$m_v \approx \rho \int_0^{+\infty} \frac{S_{k-1}(1)}{k(2\pi)^{k/2}} \xi^{k+1} e^{-\xi^2/2} \Theta\left(\frac{\tilde{m}_u C(k)}{2} \xi^2 - 1\right) := \rho T_k\left(\tilde{m}_u C(k)\right) \tag{43}$$

where we introduced $\Theta(x)$ as the Heavyside theta. Plugging in the above expression into the equation defining $m_u^t$ we have:

$$m_u = f_u^{(k)}\left(\lambda \rho T_k\left(\tilde{m}_u C(k)\right)\right) \tag{44}$$

thus approximating for small $\rho$ the function $f_u^{(k)}$, expressing everything in terms of $m_u$ and plugging in the scaling ansatz for $m_u$ in eq. (40), we obtain the simplified SE in the large sparsity regime:

$$\tilde{m}_u = C(k) T_k\left(\tilde{m}_u C(k)\right) \tag{45}$$

We can repeat the analysis done in the previous appendix to find the thresholds in this limit. The condition defining the IT threshold $\lambda_{\text{it}}$ written in eq. (37) simplifies to:

$$\int_0^{\tilde{x}} d\tilde{q} \, T_k'\left(C_k(\tilde{x})\tilde{q}\right) \tilde{q} = \int_0^{\tilde{x}} d\tilde{q} \, T_k'\left(C_k(\tilde{x})\tilde{q}\right) C_k(\tilde{x}) T_k\left(C_k(\tilde{x})\tilde{q}\right) \tag{46}$$

where we defined $C_k(\tilde{x})$ as the value of the coefficient $C(k)$, solution of eq. (45) when the overlap is fixed at $m_u = \sqrt{\frac{-\rho \log \rho}{\alpha}}\tilde{x}$. This task is much easier to solve. Likewise the computation of the dynamical spinodal simplifies greatly. We need to find the minimum SNR such that eq. (45) has a non trivial solution. By introducing the auxiliary variable $y = C_k(y)\tilde{m}$ we rewrite eq. (45) as:

$$C_k^2(y) = \frac{y}{T_k(y)} \tag{47}$$

thus the minimal SNR to obtain a non trivial solution, defined by the coefficient $C_{\text{dyn}}(k)$, to obtain a non trivial solution of the equation above is given by:

$$C_{\text{dyn}}(k) = \min_{y \in \mathbb{R}^+} \sqrt{\frac{y}{T_k(y)}} \tag{48}$$

At this stage we still need to resort to numerical inspection in order to find the coefficient $C_{\text{it}}, C_{\text{dyn}}$, but we can investigate analytically the large $k$ behaviour. By considering the leading order of the function $T_k(\cdot)$, one can see that $T_k(z) \approx \Theta\left(z - \frac{2}{k+1}\right)$. By plugging in this expression into the definition of the coefficients $(C_{\text{it}}(k), C_{\text{dyn}}(k))$ in eqs. (46),(48) we obtain the following asymptotic result:

$$C_{\text{it}}(k) \approx \sqrt{\frac{4}{k+1}} \qquad C_{\text{dyn}}(k) \approx \sqrt{\frac{2}{k+1}} \tag{49}$$

thus plugging them into the scaling assumption in eq. (40) we obtain the following scaling for the thresholds:

$$\lambda_{\text{it}} \approx \sqrt{\frac{4k^2}{k+1}} \sqrt{\frac{-\rho \log \rho}{\alpha}} \qquad \lambda_{\text{dyn}} \approx \sqrt{\frac{2k^2}{k+1}} \sqrt{\frac{-\rho \log \rho}{\alpha}} \tag{50}$$

The evolution of the coefficients $C_{\text{it}}, C_{\text{dyn}}$ as a function of the number of clusters is summarized in Fig. 9. We see in Fig. 9 that the large rank expansion is quite accurate also at moderate $k$, especially for the IT threshold. The easy phase and the alg-Bayes phase merge, hence we will not analyze the distinction between $\lambda_{\text{alg}}$ and $\lambda_{\text{alg-Bayes}}$ in this regime.

## D  Details on numerical simulations

We discuss in this section the details behind the numerical simulations presented in Sec. 4. The code is available at https://github.com/lucpoisson/SubspaceClustering. First we stress an important point on the convergence of low-rAMP algorithms. Increasing the sparsity of the problem, i.e. decreasing $\rho$, the convergence of AMP becomes more difficult. In order to solve this problem is useful to

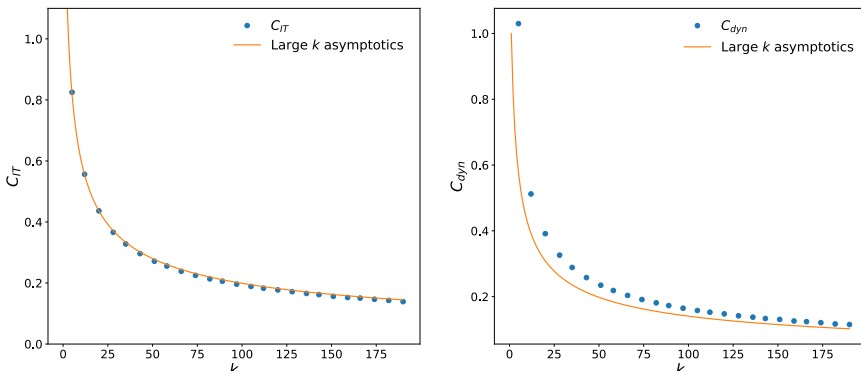

Figure 9: *Left*: Comparison of the threshold coefficients $(C_{\text{it}}, C_{\text{dyn}})$ (dots) with their high rank asymptotic expression (solid line).

---

**Algorithm 2** low-rAMP with damping

---

**Input:** Data $\mathrm{X} \in \mathbb{R}^{d \times n}$
Initialize $\hat{\boldsymbol{v}}_i^{t=0}, \hat{\boldsymbol{u}}_\nu^{t=0} \sim \mathcal{N}(\mathbf{0}_k, \epsilon\mathrm{I}_k), \hat{\sigma}_{u,\nu}^{t=0} = 0_{k \times k}, \hat{\sigma}_{v,i}^{t=0} = 0_{k \times k}$.
**for** $t \leq t_{\max}$ **do**

$\quad \mathrm{A}_u^{\text{tmp}} = \frac{\lambda}{s}\left(\hat{\mathrm{U}}^t\right)^\top \hat{\mathrm{U}}, \qquad A_v^{\text{tmp}} = \frac{\lambda}{s}\left(\hat{\mathrm{V}}^t\right)^\top \hat{\mathrm{V}}$

$\quad \mathrm{B}_v^{\text{tmp}} = \sqrt{\frac{\lambda}{s}}X\hat{\mathrm{U}}^t - \frac{\lambda}{s}\sum_{\nu=1}^n \sigma_{u,\nu}^t \hat{\mathrm{V}}^{t-1}, \quad \mathrm{B}_u^{\text{tmp}} = \sqrt{\frac{\lambda}{s}}X^\top V - \frac{\lambda}{s}\sum_{i=1}^d \sigma_{v,i}^t \hat{\mathrm{U}}^{t-1}$

$\quad$ Damping step with damping coefficient $\gamma$:
$\quad \mathrm{A}_u^t = (1-\gamma)A_u^{\text{tmp}} + \gamma A_u^{t-1} \qquad \mathrm{A}_v^t = (1-\gamma)A_v^{\text{tmp}} + \gamma A_v^{t-1}$
$\quad \mathrm{B}_u^t = (1-\gamma)B_u^{\text{tmp}} + \gamma B_u^{t-1} \qquad \mathrm{B}_v^t = (1-\gamma)B_v^{\text{tmp}} + \gamma B_v^{t-1}$
$\quad$ Take $\{\boldsymbol{b}_{v,i}^t \in \mathbb{R}^k\}_{i=1}^d, \{\boldsymbol{b}_{u,\nu}^t \in \mathbb{R}^k\}_{\nu=1}^n$ rows of $\mathrm{B}_v^t, \mathrm{B}_u^t$
$\quad \hat{\boldsymbol{v}}_i^{t+1} = \eta_v(\mathrm{A}_v^t, \boldsymbol{b}_{v,i}^t), \qquad \hat{\boldsymbol{u}}_\nu^{t+1} = \eta_u(\mathrm{A}_u^t, \boldsymbol{b}_{u,\nu}^t)$
$\quad \hat{\sigma}_{v,i}^{t+1} = \partial_{\boldsymbol{b}}\eta_v(\mathrm{A}_v^t, \boldsymbol{b}_{v,i}^t), \qquad \hat{\sigma}_{u,\nu}^{t+1} = \partial_{\boldsymbol{b}}\eta_u(\mathrm{A}_u^t, \boldsymbol{b}_{v,\nu}^t)$
$\quad$ Here $\hat{\mathrm{U}}^t \in \mathbb{R}^{n \times k}, \hat{\mathrm{V}}^t \in \mathbb{R}^{d \times k}, \mathrm{B}_u^t \in \mathbb{R}^{n \times k}, \mathrm{B}_v^t \in \mathbb{R}^{d \times k}, \mathrm{A}_u^t \in \mathbb{R}^{k \times k}, \mathrm{A}_v^t \in \mathbb{R}^{k \times k}$
**end for**
**Return:** Estimators $\hat{\boldsymbol{v}}_{\text{amp},i}, \hat{\boldsymbol{u}}_{\text{amp},\nu} \in \mathbb{R}^k, \hat{\sigma}_{u,\nu}, \hat{\sigma}_{v,i} \in \mathbb{R}^{k \times k}$

---

implement *damping* to stabilize the iteration, see the modified AMP in Algorithm 2. We compared the performance of AMP with different popular algorithm in the literature. The first general-purpose algorithm we considered for subspace clustering is a modification of the sparse PCA algorithm (SPCA). Let us consider a data matrix $Y \in \mathbb{R}^{n \times d}$, where as in our notation $n$ is the number of samples and $d$ is the feature dimension. In the SPCA problem, the statistician wants to find directions in the space which maximize the variance of our dataset by constraining the cardinality of the new basis vectors. In vanilla PCA instead we try to find directions, called principal components $\{\hat{\boldsymbol{e}}_m\}_{m=1}^d$, which maximize the variance not caring if they will be given by linear combination of all the features of our problem: $\hat{\boldsymbol{e}}_m = \sum_{i=1}^d \alpha_i^{(m)} \boldsymbol{e}_i$, where we called $\{\boldsymbol{e}_i\}_{i=1}^d$ the canonical basis vectors. In SPCA we want that some of the coefficients $\alpha_i^{(m)}$ (called "loadings" in the literature) to be zero, favouring interpretability of the optimal estimator. By formulating in a variational way the problem the sparsity of the estimator is enhanced using LASSO regularization. We write the pseudocode for the program we used in the two-class subspace clustering problem in Algorithm. 3. The unregularized problem, i.e. $\Gamma = 0$, is equivalent to vanilla PCA. The comparison of the performances of the two spectral algorithms has been done in Fig. 2 and we have a clear advantage in imposing the cardinality constraint as the sparsity level increase. We considered in the sub-extensive sparsity regime in Sec. 6 the Diagonal Thresholding algorithm (DT). The main idea is to search for spatial directions with the

---

**Algorithm 3** SPCA

---

**Input:** Data $Y \in \mathbb{R}^{n \times d}$
Initialize $\Delta_{\text{sparsity}} = 1, \Gamma = 10^{-3}$
**while** $|\Delta_{\text{sparsity}}| \geq 1$ **do**
    Solve variational problem: $(\hat{C}, \hat{D}) = \underset{C \in \mathbb{R}^n, D \in \mathbb{R}^d}{\arg\min} \{\|Y - CD^T\|_F + \Gamma \|D\|_1\}$
    Compute first sparse principal component D
    Compute the estimated sparsity $\hat{s} = \sum_{i=1}^{d}(1 - \delta_{\hat{v}_i, 0})$
    Compute sparsity mismatch $\Delta_{\text{sparsity}} = \rho d - \hat{s}$
    If $\Delta_{\text{sparsity}} < 0$ decrease $\Gamma$, otherwise increase it
**end while**
Project the data matrix onto the first sparse principal component: $P = YD \in \mathbb{R}^n$
Compute cluster membership assignment: $\hat{U} = \text{sign}(P)$
**Return:** $\text{MSE}(\hat{U})$

---

---

**Algorithm 4** Diagonal Thresholding

---

**Input:** Data $Y \in \mathbb{R}^{n \times d}$
Compute the sample covariance matrix $\hat{K} = \frac{1}{n} \sum_{\nu=1}^{n} \boldsymbol{y}_\nu \boldsymbol{y}_\nu^\top$
Find the directions with the $s$ largest variance, with $s = \lfloor \rho d \rfloor$.
Call the subset of indices corresponding to the directions above $\mathcal{S}$.
Create $\tilde{K}$:

$$\tilde{K}_{ij} = \begin{cases} \hat{K}_{ij} & \text{if} \quad (i, j) \in \mathcal{S} \\ 0 & \text{otherwise} \end{cases}$$

Compute the largest eigenvector of the thresholded matrix $\tilde{K}$ and call it $\hat{\boldsymbol{v}}$.
Project the data matrix onto the first sparse principal component: $P = Y\hat{\boldsymbol{v}} \in \mathbb{R}^n$
Compute cluster membership assignment: $\hat{\boldsymbol{u}} = \text{sign}(P)$
**Return:** $\text{MSE}(\hat{\boldsymbol{u}})$

---

largest variance, and threshold the sample covariance matrix accordingly, hence the name Diagonal Thresholding. The pseudocode for the algorithm we used in the two-classes subspace clustering is given in Algorithm. 4.