# OpenReview forum: "Subspace clustering in high-dimensions: Phase transitions & Statistical-to-Computational gap"
_NeurIPS.cc/2022/Conference — NeurIPS 2022 Accept_

### Official Review · Reviewer_Szix · 2022-07-09

**Rating:** 8
**Confidence:** 3
**Soundness:** 4 excellent
**Presentation:** 4 excellent
**Contribution:** 4 excellent

**Summary:**

This paper considers a subspace clustering problem as a low-rank matrix factorization problem. The performance of the Bayes-optimal estimator is characterized by the high-dimensional limit. Clearly, the computationally hard phase, easy phase, and impossible phase are provided via an information-theoretical threshold and an algorithmic threshold. Authors also show that the statistical-to-computational gap will grow with both the sparsity and the rank.

**Questions:**

1. Please briefly explain the meaning of the indicator vector in Eq.(3).

2. In the impossible phase, the Bayes-optimal MMSE is not better than a random guess. Does this statement hold only for the two-clusters problem? or general mixture case also applies.

3. Can the results in this paper apply to more general right-unitarily-invariant noise matrices? Recently, there are some new AMP-type algorithms such as OAMP/VAMP, CAMP, and MAMP that solve more general signal recovery problems with right-unitarily-invariant matrices. The authors may consider extending their work to right-unitarily-invariant matrices.

[OAMP] J. Ma and L. Ping, “Orthogonal AMP,” IEEE Access, vol. 5, pp. 2020–2033, 2017.
[VAMP] S. Rangan, P. Schniter, and A. Fletcher, “Vector approximate message passing,” IEEE Trans. Inf. Theory, vol. 65, no. 10, pp. 6664-6684, Oct. 2019.
[CAMP] K. Takeuchi, “Bayes-optimal convolutional AMP,” IEEE Trans. Inf. Theory, vol. 67, no. 7, pp. 4405-4428, July 2021.
[MAMP] L. Liu, S. Huang, and B. M. Kurkoski, “Memory AMP,” IEEE Trans. Inf. Theory, 20222, early access.

**Limitations:**

The results in this paper are limited to IID Gaussian noise matrices.

**Strengths And Weaknesses:**

Strengths：

1. This paper provides an exact asymptotic characterization of the MMSE in the high dimensional limit.

2. In the large sparsity regime, the authors uncover a large statistical-to-computational gap as the sparsity level grows and unveiled the existence of a computationally hard phase.

3. It shows that the SNR threshold below which recovery is statistically impossible.

4. The results in this paper solve an apparent contradiction due to the scaling assumption of the sparsity level with the dimension of the features.

5. The findings of this paper are verified by algorithms for subspace clustering such as sparse principal component analysis and diagonal thresholding.

Weaknesses

This paper only works on a low-rank matrix factorization problem with IID Gaussian noise matrices, which limits the applications of the results in this paper to more general right-unitarily-invariant noise matrices.

---

> ### Author Response · Authors · 2022-08-02
> **Reply to Reviewer Szix**
>
> We thank the referee for her/his detailed comments and for the appreciation of our work. We are particularly grateful for her/his critical reading, the suggestion for the open direction to explore is a valuable comment. We address her/his questions in the answer below.
>
> **Reply to points in "Questions"**
> - *Please briefly explain the meaning of the indicator vector in Eq.(3)*
>
> The indicator vector in eq. (3) $\bf{u}_{c}^{\nu}$ mathematically encodes that datapoint $\bf{x}_\nu$, for some $ \nu \in [n]$, is associated to cluster $c \in \mathcal{C}$. Its mathematical form, assumed without loss of generality, is just a convenient encoding for the theoretical analysis.
>
> - *In the impossible phase, the Bayes-optimal MMSE is not better than a random guess. Does this statement hold only for the two-clusters problem? or general mixture case also applies.*
>
>
> The statement is valid in general for any number of classes $k$. In Main result 1 we characterize the MMSE for any number of clusters $k$.
>
> - *Can the results in this paper apply to more general right-unitarily-invariant noise matrices? Recently, there are some new AMP-type algorithms such as OAMP/VAMP, CAMP, and MAMP that solve more general signal recovery problems with right-unitarily-invariant matrices. The authors may consider extending their work to right-unitarily-invariant matrices.*
>
> We agree with the reviewer that the Gaussian assumption is one of the limitations of our work. Indeed, a natural idea to go beyond this limitation is to consider other flavours of message passing algorithms, such as VAMP for rotationally invariant matrices. We thank the referee for this interesting suggestion, this is definitely an open direction to explore. We added in the revised "Conclusion" section a comment about this interesting (and other) open directions.

---

> > ### Comment · Reviewer_Szix · 2022-08-10
> > **Thank you for the reply**
> >
> > All of my concerns have been addressed by the authors. Based on the comments of the other reviewers and the authors' responses. I'd like to keep the same rating as in my previous review: 8: Strong Accept.

---

### Official Review · Reviewer_u83T · 2022-07-11

**Rating:** 5
**Confidence:** 4
**Soundness:** 4 excellent
**Presentation:** 2 fair
**Contribution:** 2 fair

**Summary:**

The authors consider a clustering problem where the cluster means are sparse vectors and the noise is Gaussian. They prove the information-theoretic threshold in terms of the signal-to-noise ratio (SNR) under which the clustering is impossible statistically. They also analyze a performance of an approximate message passing (AMP) algorithm for the problem. The difference between the informatin-theoretic transition and the algorithmic transition (based on the AMP algorithm) is discussed.

**Questions:**

I would like to see more about the originality and the significance of the work. In the current writing, it gives impression that the current work is merely an interpretation of existing results [3-5, 14]. The model should be explained in further detail. For example, what is the 's-sparse vector' in the first paragraph in page 2?

**Limitations:**

Yes.

**Strengths And Weaknesses:**

Strengths: The authors showed a statistical-to-computational gap for the large (but not very high) sparsity regime where the density of non-zero elements in the cluster mean is of order 1. They provide a detailed analysis between their results and the existing results for the very high sparse regime.

Weaknesses: The originality is somewhat limited as main theoretical results are based on previous results. The writing is not entirely clear.

---

> ### Author Response · Authors · 2022-08-02
> **Reply to Reviewer u83T**
>
> We thank the reviewer for the positive comments and for the interest in our work. We address each of her/his questions below.
>
> **Reply to points in "Questions"**
>
> - *[...] For example, what is the ’s-sparse vector’ in the first paragraph in page 2?*
>
> The presence of a typo in that line could have led to a confusion, and we have now corrected it the revised version. We refer to the cluster means $\bf{\mu}_c$ as the "s-sparse vector". The number of non-zero components in the centroids $s$ will be determined by the parameter $1-\rho$, which we refer to as "the sparsity level". On average the centroids will have only $s$ components different from zero, hence the name $s-$sparse vectors.
>
> - *I would like to see more about the originality and the significance of the work. In the current writing, it gives impression that the current work is merely an interpretation of existing results [3-5, 14]. The model should be explained in further detail*
>
> The starting point of our work is to map the subspace clustering problem to low-rank matrix factorization. This is a fundamental problem in statistics with a rich literature. Indeed, in our work we leverage this connection to characterize the statistical-to-computational gaps of subsplace clustering. However, we believe our work is not a "simple interpretation of existing results". First, the results derived are novel, and provide original insight into the computational hardness of subspace clustering, an important problem of interest. Second, the derivation of the sharp thresholds from eqs. (8-12) is technically involved, as the subspace clustering problem combine two hard factors: sparsity and multi-class structure. For this reason, the analysis required a series of new insights to make the equations amenable to an analytical treatment.

---

> > ### Comment · Reviewer_u83T · 2022-08-10
> > **Thank you for the response**
> >
> > Thank you for the response.

---

### Official Review · Reviewer_2rXD · 2022-07-11

**Rating:** 5
**Confidence:** 4
**Soundness:** 3 good
**Presentation:** 3 good
**Contribution:** 2 fair

**Summary:**

This paper studies the performance of subspace clustering in a high-dimensional Gaussian mixture model. The cluster centers are assumed to be sparse. The problem is equivalent to one of low-rank matrix estimation. Using this connections, the information-theoretic limit for this problem is derived and compared with the performance of AMP (generally thought to be the best among poly-time algorithms for this problem).  The results are specialized to the setting of sparsity ratio approaching one, and the statistical-computational gap is found to be of the order of $\sqrt{\rho}\log(1/\rho)$.

**Questions:**

-- Could the authors explain how the subspace clustering problem is exactly equivalent to the Eq. (4)?

-- Does the equivalence of subspace clustering to low-rank estimation hold even in the unbalanced case, i.e., when $p_c$ does not necessarily equal $1/k$? Analyzing this general case and exploring the effect of unbalancedness would be one way to increase the novelty of the contribution.

-- Abstract and l.107: could you explain what subextensive means in this context? (is it related to sublinear?)

-- Typos: l.85: "analysis" misspelled;  l.128: unpractical --> impractical; l.137-147: I think the references to Eq. (38) should be to Eq. (9)? l. 162, 172 etc. : I think Algorithm 3 refers to Algorithm 1? l.195: "investigate" misspelled.




**Limitations:**

Connections and references to prior work are good overall, but it would be nice to have at least a brief discussion of open questions and generalizations.

**Strengths And Weaknesses:**

The paper is generally well written and pleasant to read. Having exact, computable expressions for the asymptotic MMSE as well as the asymptotic overlap of AMP that are validated by numerical simuations is welcome. The scaling behavior in the large sparsity regime and the characterization of the stat-comp gap in this regime is nice.

The main weakness of the paper is novelty. As the authors acknowledge, all the results in the paper are obtained by evaluating known results for this particular model. In particular, the information-theoretic limits for low-rank matrix estimation (Theoretical result 1) have been derived in a series of  papers including [2,3,4,14], and a perusal of the supplementary material did not reveal any technical challenges in evaluating the known formulas for the subspace clustering model studied here. Similarly, the AMP and the state evolution characterization of its perforamance (Theoretical result 2) are also well known from previous works. Comparing the the IT limits with the AMP overlap just involves running the state evolution equations from two distinct initial conditions -- perfect initialization to obtain the IT limit, and uninformative initialization to obtain the AMP overlap. The behavior in the high sparsity regime is novel and a nice result, but the overall novelty of the paper still falls short.

---

> ### Author Response · Authors · 2022-08-02
> **Reply to Reviewer 2rXD 1/2**
>
> We thank the reviewer for his/her comments that help us see how to improve the presentation of our results. We address more in details some points in the answer below.
>
> **Reply to points in "Weaknesses"**
> - *The main weakness of the paper is novelty. As the authors acknowledge, all the results in the paper are obtained by evaluating known results for this particular model. In particular, the information-theoretic limits for low-rank matrix estimation (Theoretical result 1) have been derived in a series of papers including [2,3,4,14], and a perusal of the supplementary material did not reveal any technical challenges in evaluating the known formulas for the subspace clustering model studied here.*
>
> We politely disagree with the reviewer that our results are a mere "formula evaluations". Indeed, the starting point of our analysis is to map the subspace clustering problem to a low-rank matrix factorization problem, a classical problem in high-dimensional statistics with a rich literature. However, the subspace clustering problem combines two hard technical problems: sparsity and multi-class structure, and obtaining the thresholds from the generic formulas available involves a considerable tour de force, such as finding the right reduction of the equations and closing them on an ansatz for the scaling at large sparsity. We detail these technical challenges also in the reply to *Reviewer U268*.
>
> - *[...] Similarly, the AMP and the state evolution characterization of its perforamance (Theoretical result 2) are also well known from previous works. Comparing the the IT limits with the AMP overlap just involves running the state evolution equations from two distinct initial conditions -- perfect initialization to obtain the IT limit, and uninformative initialization to obtain the AMP overlap. The behavior in the high sparsity regime is novel and a nice result, but the overall novelty of the paper still falls short.*
>
> The  IT threshold cannot be computed by simply running the state evolution equations from uninformed and informed initializations. In fact, a close inspection of Fig. 2b, shows that *only after* the IT threshold (dashed light blue line) the MMSE is achieved by the informed initialization (solid orange line). Indeed, the exact computation of this threshold requires evaluating the free energy potential, which is technically more involved than running state evolution (see App. B for the details).
>
> We thank the referee for his/her positive comments on the results in the high sparsity regime, although we politely disagree, for all the reasons above, that the novelty of this paper falls short.

---

> > ### Author Response · Authors · 2022-08-02
> > **Reply to Reviewer 2rXD 2/2**
> >
> > **Reply to points in "Questions"**
> >
> > - Could the authors explain how the subspace clustering problem is exactly equivalent to the Eq. (4)?
> >
> > In the subspace clustering problem we are interested to find the relevant features in our dataset.  In order to study that, we consider the data model in eq. (1), in which we have a low-dimensional subspace of relevant features, determined by the non-zero components of the cluster means, embedded in a much higher dimensional space, defined by the acquisition space $d$. Mapping eq. (1) (subspace clustering) to eq. (4) (low-rank matrix factorization) requires finding the appropriate parametrization, given here by the vectors $(\bf{v} _i, \bf{u}_c^{\nu} )$ with ${i \in [d], \nu \in [n]}$.
> >
> > - *Does the equivalence of subspace clustering to low-rank estimation hold even in the unbalanced case, i.e., when $p_c$ does not necessarily equal ? Analyzing this general case and exploring the effect of unbalancedness would be one way to increase the novelty of the contribution.*
> >
> >  Although it would be possible to treat the unbalanced case exploiting the low-rank estimation mapping, moving away from the balanced case would end up in considering an easier problem. Since the main goal of this work is to shed light on the statistical-to-computational gap and presence of hard phases in the subspace clustering problem, we believe the balanced case is the most relevant regime. However, we completely agree with the reviewer that the extension to the unbalanced case is an interesting open direction and we added a comment about it in the conclusions.
> >
> > - *Abstract and l.107: could you explain what subextensive means in this context? (is it related to sublinear?)*
> >
> > The referee is correct and it is indeed related to sublinear in this context, with respect to the dimension $d$. In our setting we consider the density of non-zero elements $s$ in the cluster means to scale with the dimension $d$, which is taken to infinity. Hence, we have $\rho = \frac{s}{d} = O(1)$. In the sub-extensive (or sub-linear) regime we allow the number of non-zero component $s$ to scale sub-linearly, hence $\rho = \frac{s}{d} = o(1)$.
> >
> > - *I think the references to Eq. (38) should be to Eq. (9) [....] I think Algorithm 3 refers to Algorithm 1?*
> >
> > We thank the reviewer for the suggestions and for pointing out typos. We welcomed all of the suggestions and corrected the typos in the revised version. We confirm that his/her interpretation of this sentence was correct: in the new version eq. (38) is replaced by eq. (9), and Algorithm 3 by Algorithm. 1.
> >
> > **Reply to points in "Limitations"**
> > - *Connections and references to prior work are good overall, but it would be nice to have at least a brief discussion of open questions and generalizations.*
> >
> > We thank the referee for this comment. The mapping of subspace clustering to low-rank matrix factorization is flexible and opens up interesting open directions. We added a comment in the conclusions, clarifying the limitations of this work and highlighting prospective future directions.

---

> > ### Comment · Reviewer_2rXD · 2022-08-09
> > **Reply to authors**
> >
> > Thanks very much for your answers and clarifications. It is indeed interesting that the IT-threshold does not coincide with the one corresponding to AMP with fully informed initialization. The subtleties in the phase diagram easy to miss as much of the interesting discussion on the thresholds is relegated to the supplementary material.
> >
> > I have another question regarding computation of the IT threshold. Is this done via Theorem 1, using a brute force evaluation of the minimizer in (9)? It would nice to have some discussion of this optimzation as minimizing over the space of all $k \times k$ matrices quickly becomes infeasbile as $k$ grows.
> >
> > I am now more persuaded about the novelty of the contribution, but the paper needs to be rewritten to clearly highlight the novel aspects and implications in the main text.  Thanks again for your helpful responses.

---

> > > ### Author Response · Authors · 2022-08-09
> > > **Thank you for your reply to our rebuttal**
> > >
> > > We thank the reviewer for the answer to our rebuttal and we are happy that we could clarify most of the points. We do hear that the reviewer has still some concerns on the clarity of our work, and we address his/her question below.
> > >
> > > - *I have another question regarding computation of the IT threshold. Is this done via Theorem 1, using a brute force evaluation of the minimizer in (9)? It would nice to have some discussion of this optimzation as minimizing over the space of all kxk matrices quickly becomes infeasbile as k grows.*
> > >
> > > The computation of the IT threshold is greatly simplified thanks to the discussion in Sec. 5. We find a suitable parametrization of the $k \times k$ matrices (Mu,Mv) in eq. (19) which is invariant under the state evolution updates and reduce the number of parameters to be tracked, although we allow $k$ to be large.  The details of the computation of the IT threshold are given in Appendix B: we need to evaluate the difference of the free energy potential between the informed and uinformed fixed points and find the SNR such that this quantity is zero, see eq. (38) and Fig. 6.
> > >
> > > - *I am now more persuaded about the novelty of the contribution, but the paper needs to be rewritten to clearly highlight the novel aspects and implications in the main text.*
> > >
> > > We thank the reviewer for the remarks on the clarity of our work. We will make sure to clearly state what are the novel implications of our work in the revised version. We are aware that if the paper is accepted this discussion will be made public and we would not be making it if we were not to stand by it.

---

### Official Review · Reviewer_a17U · 2022-07-11

**Rating:** 6
**Confidence:** 3
**Soundness:** 3 good
**Presentation:** 3 good
**Contribution:** 3 good

**Summary:**

This paper studies statistical and computational limits of subspace clustering, focusing on high-dimensional Gaussian mixture model with sparse centroids. The paper considers the regime where both the fraction of non-zero components of the cluster means and the ratio between the number of samples and the dimension are fixed (which differs from previous literature). The information-theoretic threshold (below which reconstruction is statistically impossible) and algorithmic limitations (in terms of an approximate message passing algorithm) are obtained, which indicate a statistical-to-computational gap. The results are illustrated and compared with previous works through simulations.





**Questions:**

I have the following comments on details:
(1) Line 12 in the Abstract: "require" might be "requires"?
(2) Line 36: it would be nice if a definition of "s-sparse vectors" can be given here: is it in terms of expectation or deterministic?
(3) Line 53: the second "is" might be "in"?
(4) Line 84: "arise" might be "arises"?
(5) Line 138: "allow" might be "allows"?
(6) Line 177: is it "closest to"?
(7) Line 195: "invextigate" might be "investigate"?
(8) Line 204: is it "have access to"?

**Limitations:**

The limitations of the work are already discussed in the paper.

**Strengths And Weaknesses:**

The analysis of statistical and computational limits and the resulting phase diagram in this paper is an interesting contribution to the study of high-dimensional Gaussian mixture models and subspace clustering. The regime that the paper focuses on (extensive sparsity: the fraction of non-zero components of the cluster means is fixed) is different from that in the previous literature (where the fraction of non-zero components of the cluster means goes to 0). Many of the theoretical results are adapted from existing analysis on statistical and computational limits of low-rank matrix factorization, although the idea that statistical and computational limits of Gaussian mixture model can be connected to low rank matrix factorization is new. The ideas and results are clearly presented in the paper, and the phase diagram and simulation studies give good illustrations to the statistical-and-computational gap.

---

> ### Author Response · Authors · 2022-08-02
> **Reply to Reviewer a17U**
>
> We thank the referee for the appreciation of our work, we also like very much the illustrative power of the phase diagram in Fig. 1. We thank the reviewer for the list of suggestions and typos, we welcome all of them, and are corrected in the revised version. We address his/her question below.
>
> **Reply to points in "Questions"**
> - *It would be nice if a definition of "s-sparse vectors" can be given here: is it in terms of expectation or deterministic?*
>
> Thank you for the valuable comment, indeed the definition of "s-sparse vector" is given in terms of expectation. We added a comment in the revised version to clarify this point at line 51. Thanks to the assumption of the Gauss-Bernoulli distribution in eq. (2) with parameter $\rho = \frac{s}{d}$, the cluster means $\bf{\mu}_c $ will have, in expectation, a number of non-zero components equal to $s$.

---

### Official Review · Reviewer_U268 · 2022-07-16

**Rating:** 4
**Confidence:** 3
**Soundness:** 2 fair
**Presentation:** 2 fair
**Contribution:** 3 good

**Summary:**

This work explores statistical-computational gaps in the problem of obtaining non-trivial correlation (sometimes referred to as `weak recovery') with the cluster labels of a balanced mixture of isotropic Gaussian vectors that have sparse means drawn from a Gaussian-Bernoulli prior. Via a reduction to a low-rank matrix factorization problem, explicit formulas are derived for the asymptotic mean-squared error of the Bayes estimator, which is statistically optimal for this problem although likely hard to compute. On the algorithmic side, approximate message passing is proposed and an exact asymptotic formula for its mean-squared error is derived. Information-theoretic and algorithmic thresholds are described, and for large enough sparsity levels, these two thresholds differ. Numerical experiments explore what happens when the sparsity level is sub-extensive and show that the phase transitions have a different behavior in this regime.

**Questions:**

**1.** Is it possible to understand intuitively why the algorithmic threshold does not depend on $\rho$?

**2.** "The state evolution equations (17) coincide exactly with running gradient descent
175 on the potential defined in eq. (40)!" Is this connection known for other problems? Why is this observation crucial for the results here?

**Limitations:**

**1.** I agree with the authors' assessment that there are no apparent potential negative societal impacts.

**2.** The weak recovery problem studied here is primarily of theoretical interest, and it is not clear if the AMP algorithm is useful for non-Gaussian problems. So practical impact may be limited.

**Strengths And Weaknesses:**

# Strengths

- *Significance/Quality*: The theorems give a roughly complete understanding of the weak recovery problem, from an information-theoretic and algorithmic point of view, in the linear-sparsity, linear-dimension regime. In particular the algorithmic threshold is identified with a sharp constant factor.

- *Clarity*: The simulations for the large-sparsity regime and discussion are helpful for understanding the complex behavior of this problem as the sparsity varies.

# Weaknesses

- *Originality*: Main Result 1 relies on known formulas for low-rank matrix factorization. It is not clearly explained what are the major technical challenges, if any, in obtaining this result.

- *Clarity*: The community labels in (3) and the model (4) are such that the $\mathbb{E}X$ does not have sparse columns if $k$ is small. For this reason, I feel the paper is more about the large $k$ version of the sparse clustering problem.

**Edit 08/19:** After discussion with authors, the previous point is resolved.

- *Clarity*: From the main text alone, it is unclear how the information-theoretic threshold is obtained. The formula of the MSE is difficult to interpret, so I can't see if the threshold is a consequence of this. Some further explanation is needed here.

- *Quality/Clarity*: It is difficult to assess the rigour of both main results, especially Main Result 2. This is because the appendix is not organized in a conventional way with a clearly demarcated proof of Main Results 1 and 2. For Main Result 2, I do not see in the Appendix any explanation of how the asymptotic algorithmic MSE is computed. I only see plots rather than arguments. I also do not see any derivation of the Bayes-optimal MSE or reference to known (rigorous) formulas.

# Minor issues

- Line 39: Equation (2) defines vectors that are (i) standard Gaussian with probability $\rho$ OR (ii) the zero vector with probability $1 - \rho$. I think what is meant is for there to be a random subset of zero entries, with the rest of the entries being Gaussian.

- Main Theorem 1: Please take a careful proofread over this. Here $\mathbf{v^*}, \mathbf{u^*},$ and $\mathbf{w}$ have not been defined in (10). Also $Z_u$ does not appear in (10).

- Consider making the boundaries bolder in Figure 1. Also I found the color of $\lambda_{it}$ to be hard on the eyes

- Line 70: Typo "statitiscal"

- Line 85: Typo "analyis"

- Line 91: Typo "Statistics", change to "statistics"

- Line 111: Change to "In particular, [10] conjectured and [11] proved..."`

- Line 143-144: This comment is hard to understand because (38) is in the Appendix, and then I'm having trouble seeing the connection to (6).

- Line 195: Typo "Invextigate"

- Line 261: Replace "Despite of this fact" with "Despite this fact"

# Summary of score

My score is due to concerns mostly about the rigor and partially about the novelty of this submission. I also feel there is a lack of clarity in explaining how the main results are obtained.

# Update of score 08/19

The authors' rebuttal addressed my concerns about rigor and somewhat about novelty. I agree with other reviewers that the strengths and technical challenges of this paper are not highlighted enough in the main text. I also think further clarity is needed on the level of rigor and the asymptotic regime to which the results apply (which seems to be for k growing large and rho going to 0). I have raised my overall score from 3 to 4 because further serious revision is needed. I have also upgraded the soundness from 1 to 2.

---

> ### Author Response · Authors · 2022-08-02
> **Reply to Reviewer U268 1/2**
>
> We thank the referee for her/his comments  that help us see where the discussion of our paper should be improved. Below, we address the concerns and questions raised by the reviewer.
>
> **Reply to points in "Weaknesses"**
>
>  -  *Originality: Main Result 1 relies on known formulas for low-rank matrix factorization. It is not clearly explained what are the major technical challenges, if any, in obtaining this result.*
>
> The main goal of our work is to analytically characterize the statistical-to-computational gap in subspace clustering at large sparsity. Our first result is to identify the connection between the problem of interest (subspace clustering) and a low-rank matrix factorization problem. Indeed, this connection allows us to leverage the rich literature from this field, and in particular a set of very generic formulas for the MMSE. However, using these formulas for subspace clustering and computing the respective thresholds analytically is far from trivial and consitute a considerable technical challenge.
>
> The two major difficulties are (details are given in Secs. 5 and 6): (a). Find a suitable parametrization of the $k\times k$ matrices $(M_u,M_v)$ in eq. (19) which is invariant under the state evolution updates and reduce the number of parameters to be tracked; (b) Find the right scaling ansatz in the high-sparsity limit $\rho \to 0^{+}$ which allow us to close the equations on amenable quantities. Thanks to these two non-trivial steps, we are able to find closed expressions for the relevant thresholds which are plotted in the phase diagram Fig. 1, and analytically characterize the statistical-to-computational gap in the large sparsity regime of interest.
>
> We definitely agree with the reviewer that these challenges were not stressed enough, and we will add a discussion in the revised version.
>
> - *Clarity: The community labels in (3) and the model (4) are such that the E X does not have sparse columns if k is small. For this reason, I feel the paper is more about the large k version of the sparse clustering problem.*
>
> The sparsity of the matrix $\mathbb{E} \left[X \right]$ is associated only to the sparsity level in the cluster means ($1-\rho$) and not with the parameter $k$, therefore the model is well defined for any number of cluster $k \geq 2$.
>
> - *Clarity: From the main text alone, it is unclear how the information-theoretic threshold is obtained.*
>
> The detailed discussion of how to compute the information-theoretic threshold is given in App. B, and in particular in eq. (38). The key idea is to compute the difference in the free energy potential $\Phi_{\text{rs}}$ between the "informative" and "uninformative" fixed points, and to search for the SNR for which the difference is zero. This is illustrated in Figs. (5,7).
>
> - *Clarity: [...] The formula of the MSE is difficult to interpret, so I can’t see if the threshold is a consequence of this. Some further explanation is needed here*
>
> We completely agree with the reviewer that it is not immediate to get the thresholds from the formulas, which in full generality are quite involved. As we discussed above, although eqs. (8-12) is all you need at a first glance, it is hard to approximate them numerically with a good precision and technically challenging to compute the thresholds analytically from eqs. (8-12). The computation takes a few pages, and for this reason we detail it in App. B.
>
> - *Quality/Clarity: It is difficult to assess the rigour of both main results, especially Main Result 2. This is because the appendix is not organized in a conventional way with a clearly demarcated proof of Main Results 1 and 2.*
>
> Once the mapping to the low-rank matrix factorization is done, closed-formula characterizing the MMSE has been proven in a series of works [3-5,14]. The second main result comes from the general rigorous connection between AMP algorithm and its associated SE [31].
> Note, however, that we choose to write "theoretical results" instead of "theorem". This is because,  when we evaluated our closed-form formulas, we did not take all the necessary precautions to claim full rigor,  e.g. prove formally that the minimum was unique (though we checked all these both analytically and numerically).
>
> - *[...] For Main Result 2, I do not see in the Appendix any explanation of how the asymptotic algorithmic MSE is computed. I only see plots rather than arguments*
>
> The asymptotic algorithmic MSE is computed by iterating the SE equation defined in eqs. (16,17) until convergence. We plug the fixed point of the iteration into eq. (18) in order to find the asymptotic performance.
>
> - *Quality/Clarity: [...] I also do not see any derivation of the Bayes-optimal MSE or reference to known (rigorous) formulas*
>
> The formula in eq. (8) of the MMSE, is derived directly from the definition of the MSE in eq. (5) by exploiting the prior distribution on the encoding labels $P_u$. The expression is rigorously proven in a series of works [3-5,14].

---

> > ### Author Response · Authors · 2022-08-02
> > **Reply to Reviewer U268 2/2**
> >
> > **Reply to points in "Minor issues"**
> >
> >  We thank the referee for the list of suggestions and typos in this section, we welcome all of them, and they have been fixed in the revised version. We make sure that there is no ambiguity left, by answering in detail to some comments in the answer below.
> >
> > - *Line 143-144: this comment is hard to understand because (38) is in the Appendix, and then I'm having trouble seeing the connection to (6).*
> >
> > There was a hitch in the cross-referencing, the comment in line 143-144 will not create any confusion in the revised version: eq. (38) is replaced with the correct reference, i.e. eq. (9), and the minimisation problems in eq. (6) is strongly different from the one in eq. (9) since the latter involves high dimensional quantities intractable in the limit considered $n,d \to \infty$.
> >
> > - *Equation (2) defines vectors that are (i) standard Gaussian with probability $\rho$ OR (ii) the zero vector with probability $1-\rho$. I think what is meant is for there to be a random subset of zero entries, with the rest of the entries being Gaussian.*
> >
> > Indeed, there was a typo in the definition of the vector $\bf{v}_i$. This is corrected in the revised version, and there should be no ambiguity left: the vector $\bf{v}_i \in \mathbb{R}^k$ is the selector of the relevant features in the mixtures. For a given direction $i \in [d]$, the vector $\bf{v}_i$ is the null vector with probability $1-\rho$, meaning that feature $i$ is irrelevant as all the centroids have zero component along that direction.
> >
> > **Reply to points in "Questions"**
> >
> >  - Is it possible to understand intuitively why the algorithmic threshold does not depend on $\rho$?
> >
> > We completely agree that this is an important and unintuitive point. We study the algorithmic threshold in Sec. 5, in particular by linearizing the SE equations around the uninformative fixed point, we found that $\lambda_{\text{alg}} = \frac{k}{\sqrt{\alpha}}$, i.e. the so-called BBP threshold [13]. Thanks to this identification we can build intuition on this result from the standpoint of Random Matrix Theory. If we work at low enough SNR ($\lambda \ll 1$), the data matrix $X$ in eq.(4) resembles effectively a Gaussian matrix, whose spectral density is close to the Wigner semicircle distribution. As we increase the SNR, the BBP transition is associated to the emergence of an informative eigenvalue, due to the low-rank part of the matrix $X$, from the "Gaussian bulk" of the spectrum. Therefore, we note that we can tune the strength of the two terms in $X$ (low-rank vs. high-rank) by varying the SNR $\lambda$. We also intuitively understand that as we decrease $\rho$, i.e. increase the sparsity of the centroids, the clustering problem becomes *statistically* easier. Nevertheless, at fixed SNR, decreasing $\rho$ does not change the relative strength of the two competing terms in the matrix $X$, hence it does not change the behaviour of the algorithmic transition.
> >
> > - *The state evolution equations (17) coincide exactly with running gradient descent 175 on the potential defined in eq. (40)!" Is this connection known for other problems? Why is this observation crucial for the results here?*
> >
> > This connection is indeed at the roots of the conjectured link between hardness and AMP. This has been explored in many references,  e.g. [7] [14], or very recently in [BAH$^+$22].
> >
> > The observation is crucial in order to analyze the phase transitions for the subspace clustering problem:
> > a) the linearization of the SE equations around the uninformative fixed point, done in Sec. 5, is equivalent to study the gradient of the potential $\Phi_{\text{rs}}$ in the origin;  b) the statistical-to-computational gap opens up as the global minima of the potential is attained far from a locally stable region around the uninformative fixed point, where the AMP algorithm get stuck (see Figs. 5 and 7).
> >
> > [BAH$^+$22]:  Afonso S. Bandeira, Ahmed El Alaoui, Samuel B. Hopkins, Tselil Schramm, Alexander S.
> > Wein, and Ilias Zadik. The franz-parisi criterion and computational trade-offs in high
> > dimensional statistics, 2022.
> >
> > **Reply to point in "Limitations"**
> > - *The weak recovery problem studied here is primarily of theoretical interest, and it is not clear if the AMP algorithm is useful for non-Gaussian problems. So practical impact may be limited*
> >
> > We agree with the referee that our work is primarily of theoretical interest. However, to the best of our knowledge this work is the first to analyse the statistical-to-computational gap of subspace clustering for extensive sparsity, and therefore we think that it is a natural choice to present the results in the Gaussian case. We agree with the reviewer that considering other distributions is an interesting direction. For instance, rotationally invariant distributions could be amenable to a similar treatment using VAMP [36]. We have added this interesting prospective direction in the "Conclusion" section of the revised manuscript

---

> > > ### Comment · Reviewer_U268 · 2022-08-08
> > > **Reply to authors**
> > >
> > > Thank you for the detailed and thoughtful response. I have a few more comments and questions below.
> > >
> > > 1. "*The sparsity of the matrix  is associated only to the sparsity level in the cluster means...*"
> > >
> > > My understanding is that $VU^T$ has columns that comprise the cluster means specified as $\mu_c$ in (1). Usually, the community labels are one-hot vectors, but here a different choice of labels is used as in (3). Can you tell me roughly what is the sparsity of the columns of $VU^T$?
> > >
> > > 2. *Regarding threshold derivations*
> > >
> > > I have taken a closer look at the Appendix. My understanding is that the thresholds stated in the theorem are derived in Appendix C based on taking limits as $\rho \to 0^+$ and $k \to \infty$. Is it true that the results only apply for a specific limiting scenario? I am not clear about the `rate' at which these limits are taken with respect to each other (eg, does $\rho k \to \infty$?). In this limit, are we still guaranteed that the cluster means (columns of $VU^T$) are sparse?
> > >
> > >
> > > 3. "*Note, however, that we choose to write "theoretical results" instead of "theorem". This is because...*"
> > >
> > > I greatly appreciate the clarity on this. A remark along these lines in the main text would help readers understand the level of rigor of this work.
> > >
> > > 4. *Regarding the level of rigor*
> > >
> > > I am fine with the level of rigor described in your response, as long as the paper is precise about what is mathematically proven and what results are shown in the calculations. For example, I would appreciate some further clarity in 2 above regarding what asymptotic regimes in $\rho, k$ your results apply to.
> > >
> > > Again, thank you for the helpful reply.

---

> > > > ### Author Response · Authors · 2022-08-09
> > > > **Thank you for your reply to our rebuttal**
> > > >
> > > > We thank the reviewer for the answer to our rebuttal and we are happy that we could clarify most of the points. We do hear that the reviewer's biggest concern remains clarity, and we address his/her question below.
> > > >
> > > > - *My understanding is that $VU^T$ has columns that comprise the cluster means specified as in (1). Usually, the community labels are one-hot vectors, but here a different choice of labels is used as in (3). Can you tell me roughly what is the sparsity of the columns of $VU^T$?*
> > > >
> > > > The reviewer is correct. While the one-hot encoding is more common, the mathematical treatment would be more cumbersome. We have exploited the parametrization freedom for the community labels to judiciously choose an encoding that simplifies the analysis.
> > > > The columns of $VU^T$ do comprise the cluster means. The number of non-zero elements in the columns of the matrix $VU^T$ is determined by the sparsity of the centroids (quantified by $\rho = \frac{s}{d}$), and therefore is equal to $s$ (on average).
> > > >
> > > > - *I have taken a closer look at the Appendix. My understanding is that the thresholds stated in the theorem are derived in Appendix C based on taking limits as $\rho \to 0$ and $k \to \infty$. Is it true that the results only apply for a specific limiting scenario? I am not clear about the `rate' at which these limits are taken with respect to each other (eg, does ?). In this limit, are we still guaranteed that the cluster means (columns of ) are sparse?*
> > > >
> > > > We thank the rewiever for the critical reading of the appendix. The mathematical derivation in order to compute the analytical scaling of the IT threshold is: a) First take the proportional high-dimensional limit $n,d,s\to\infty$ at fixed $\alpha=n/d$, $\rho=s/d$ and $k$; b) Then take the high-sparsity limit $\rho \to 0^{+}$; 3) Finally consider large $k\to\infty$ for approximating the integral appearing in eq 45.
> > > > In this limit we are still guaranteed that the columns of $VU^T$ are sparse: it will depend on the value of $\rho$ as we discussed in the first point. However, as showed in Fig 9 (left) the large $k$ approximation of eq. 45 for the IT transition is fairly accurate even at moderate $k \sim 10$.
> > > >
> > > > - *I am fine with the level of rigor described in your response, as long as the paper is precise about what is mathematically proven and what results are shown in the calculations. For example, I would appreciate some further clarity in 2 above regarding what asymptotic regimes in $\rho,k$ your results apply to.*
> > > >
> > > > We thank the reviewer for pointing this out - we agree that we might not have stressed enough which results are rigorous or not. We will make sure to clarify what results are rigorous and what are conjectured in a revised version. We are aware that if the paper is accepted this discussion will be made public and we would not be making it if we were not to stand by it.

---

> > > > > ### Comment · Reviewer_U268 · 2022-08-09
> > > > > **Re: Thank you for your reply to our rebuttal**
> > > > >
> > > > > Thanks very much for the fast and thoughtful reply.
> > > > >
> > > > > Sorry, I'm still a little confused about the sparsity. If I understand correctly, a column of $VU^T$ takes the form (up to relabeling indices) $\mu = \frac{k-1}{k} v_1 - \frac{1}{k} \sum_{i =2}^{k} v_i$ , where the $v_i$ are sampled as in (2) and have (roughly) $s = \rho d$ nonzero entries.
> > > > >
> > > > > If $k$ is large (say $k \gg \sqrt{d}$), then the second term should "average out", resulting in a mean vector $\mu$ with roughly $s = \rho d$ nonzero entries. But for smaller $k$, the supports of the $v_i$ will not entirely overlap -- so in principle the number of nonzero entries of $\mu$ could be something like $k s$.
> > > > >
> > > > > Is my understanding correct here? (Also I acknowledge that we are quite close to the author discussion deadline -- if you are not able to reply in time, and I will address this question with the other reviewers.)

---

> > > > > > ### Author Response · Authors · 2022-08-09
> > > > > > **Clarification on the setting**
> > > > > >
> > > > > > We thank the reviewer for the specific question, we hope that this (very fast) answer will help to address his/her concern on the setting of our work.
> > > > > >
> > > > > > We detail more explicitly how the matrix $VU^T$ is built: let us consider a fixed sparsity $\rho = s/d$ and that datapoint $\nu$ belongs to the first cluster , hence  the label vector is $u_{\nu} = (k-1/k, -1/k, \dots, -1/k)$. The vector $v_i$,  for all $i \in [d]$, will be sampled from a Gauss Bernoulli (GB) distribution and it will be the null vector with probability $1-\rho$ or a Gaussian vector with unit covariance with probability $\rho$.
> > > > > > Now let us consider the i-th element of the column $\nu$ of the matrix $VU^T$ , this will be equal to $v_i^T u_{\nu}$: this element will be zero with probability $(1-\rho)$ following the reasoning above on the sampling of the vector $v_i$. Repeating this reasoning for all $i \in [d]$ we have that the number of non-zero elements in the column $\nu$ of the matrix $VU^T$will be $s$ (on average).
> > > > > >
> > > > > > We hope that a practical example could help to clarify the confusion. For the sake of simplicity we consider  a low dimensional case, although the results shown in our work are valid only in high dimensions.
> > > > > > Take $(d=3,n=2,k=2)$ and fix a sparsity level of $\rho = 33$ % , meaning that “two features out of three will be irrelevant”. Now consider that our mixture is formed by two datapoints, one in the first cluster and the other in the second one.
> > > > > >
> > > > > > We sample the vectors $(v_i)_{i \in [d]}$ with a GB distribution and the resulting matrices are:
> > > > > >
> > > > > > $V = \begin{pmatrix} 0&0 \\\\ z_1&z_2 \\\\ 0&0 \end{pmatrix}$ and $U = \begin{pmatrix} 1/2 & -1/2 \\\\ -1/2 & 1/2 \end{pmatrix}$
> > > > > >
> > > > > > where  the vector $z = (z_1,z_2)$ is a unit covariance Gaussian vector.
> > > > > >
> > > > > > We considered in this simple example that the number of non-zero elements in the columns of $V$ are exactly equal to $s=1$, but in high dimensions we will have $s = \rho d$ relevant features only on average.
> > > > > >
> > > > > > We thank the reviewer for his/her critical reading and for pointing out this source of confusion, we will explain in more detail the construction above in the revised version.

---

> > > > > > > ### Comment · Reviewer_U268 · 2022-08-09
> > > > > > > **Thanks!**
> > > > > > >
> > > > > > > Thanks for the rapid reply! Your note clears up my misunderstanding. I see now that the supports for all of the *columns* of V are actually equal to each other.

---

### Meta-Review · Area_Chair_F9go · 2022-08-26

**Recommendation:** Accept
**Confidence:** Certain

**Metareview:**

The reviewers appreciate the solid theoretical results concerning statistical-computational tradeoff in subspace clustering. The exact asymptotics and clear presentation make the paper stand out. Therefore, I recommend acceptance. Meanwhile, please carefully revise the paper according to the reviews to highlight its strengths and originality.

**Award:**

No

---

### Decision · Program_Chairs · 2022-09-14

Accept